# Accelerated Sampling from Masked Diffusion Models via Entropy Bounded Unmasking

**Heli Ben-Hamu**    **Itai Gat**    **Daniel Severo**    **Niklas Nolte**    **Brian Karrer**

FAIR, Meta AI

## Abstract

Recent masked diffusion models (MDMs) have shown competitive performance compared to autoregressive models (ARMs) for language modeling. While most literature has focused on performance enhancing sampling procedures, efficient sampling from MDMs has been scarcely explored. We make the observation that often a given sequence of partially masked tokens determines the values of multiple unknown tokens deterministically, meaning that a single prediction of a masked model holds additional information unused by standard sampling procedures. Based on this observation, we introduce *EB-Sampler*, a simple drop-in replacement for existing samplers, utilizing an **E**ntropy **B**ounded unmasking procedure that dynamically unmasks multiple tokens in one function evaluation with predefined approximate error tolerance. We formulate the EB-Sampler as part of a broad family of adaptive samplers for which we provide an error analysis that motivates our algorithmic choices. EB-Sampler accelerates sampling from current state of the art MDMs by roughly 2-3x on standard coding and math reasoning benchmarks without loss in performance. We also validate the same procedure works well on smaller reasoning tasks including maze navigation and Sudoku, tasks ARMs often struggle with.

## 1 Introduction

Motivated by the success of diffusion and flow Esser et al. (2024); Polyak et al. (2024) models in continuous domains (*e.g.*, images, video), research efforts have focused on adapting these frameworks to discrete state spaces. Many of these approaches (Austin et al., 2021a; Lou et al., 2023; Sahoo et al., 2024; Campbell et al., 2024; Gat et al., 2024; Shi et al., 2024; Kitouni et al., 2024) utilize the masked modeling paradigm, thus we generally refer to these approaches as Masked Diffusion Models (MDMs). Recent large MDMs such as LLaDa (Nie et al., 2025b) and Dream (Ye et al., 2025) have shown competitive performance compared to similarly sized autoregressive models (ARMs) in the 7 billion parameter range on traditional language tasks including code, text, and mathematical reasoning benchmarks. These results open the possibility of scaled non-autoregressive generation for language using MDMs, a possible competitor to large language models (LLMs).

Abstractly, MDMs sample fixed-length sequences as discrete tokens from a vocabulary. They begin from a sequence of mask tokens and iteratively update tokens until all tokens are unmasked. Replacing a masked token with a token from the vocabulary is *unmasking*. In contrast to standard ARMs, the order in which tokens are unmasked becomes a design choice. Recent models often leverage *samplers* that surpass the performance of random order unmasking, deviating from the masked diffusion process used at training (Austin et al., 2021a). In particular, LLaDa and Dream achieve strong performance by unmasking tokens in a much more favorable order dictated by model predictions (Chang et al., 2022; Kim et al., 2025; Zheng et al., 2023).

Performance is not the only criteria by which large MDMs have become popular. Another additional important aspect is efficient computation. While MDMs cannot reuse past computation via key-value caching like ARMs due to full attention, MDMs offer an exciting alternative route to efficiency via the opportunity to sample multiple tokens simultaneously. MDM efficiency is captured by the number of function evaluations (NFE) which is directly controlled via the sampler. The default for more efficient sampling of MDMs like LLaDa and Dream is to unmask a fixed number of tokens each step. Unfortunately, generation quality degrades quickly as more unmasked tokens are sampled independently and hence this opportunity has been thus far unrealized. Developing an MDM sampler for language that simultaneously achieves good performance and efficiency is the focus of this work.

Our approach is motivated by two empirical observations. The first is the strong performance of unmasking orders determined by the model, which indicates the associated predictions aligned with that order may have low model error, i.e. match the desired optimal distribution. The second is that multiple masked tokens are often predicted with high certainty, or, equivalently, low entropy. Intuitively, masked tokens that have both low model error and low entropy can be unmasked in parallel, which we propose to exploit with our approach. By strategically unmasking such tokens, a sampler can follow high performance unmasking orders while avoiding error due to independently unmasking tokens.

We realize these intuitions and make them concrete and precise via our main contributions:

- We propose an adaptive sampler for masked diffusion models called *EB-Sampler* (short for **E**ntropy **B**ounded Sampler), that decides both which tokens to unmask and how many tokens to unmask at each step using an entropy bound that approximately limits the dependence of unmasked tokens.

- EB-Sampler is a simple drop-in replacement to existing samplers and can be directly applied to sample from existing masked diffusion models without further training.

- EB-Sampler accelerates sampling from the best performing masked diffusion models (LLaDa and Dream) by 2-3x on standard coding and math reasoning benchmarks without loss in performance. We provide Pareto fronts between efficiency and performance and conclude EB-Sampler is superior across most settings. We also validate EB-Sampler works well on smaller reasoning tasks, including maze navigation and Sudoku.

- We show EB-Sampler is a member of a broad family of adaptive multi-token samplers for masked diffusion models. Our theory explains why this family samples correctly, why it can leverage a pre-trained masked diffusion model, and provides an intuitive error decomposition into model error and joint dependence error that motivates the design of EB-Sampler.

## 2 Preliminaries

### 2.1 Notations

Consider a discrete *state space* denoted $\mathcal{S} = \mathcal{T}^d$, of sequences of length $d$ over a finite sized *vocabulary* $\mathcal{T} = [K] = \{1, 2, \ldots, K\}$. A state, $x \in \mathcal{S}$, is a sequence of elements, often called *tokens*, from the vocabulary $\mathcal{T}$, where $x = (x^1, x^2, \ldots, x^d)$ with each element $x^l \in \mathcal{T}$. We denote the set of indices of a sequence in $\mathcal{S}$ with $\mathcal{I} = [d] = \{1, 2, \ldots, d\}$. For masked modeling, we extend our vocabulary to include a mask token, $\mathtt{m}$. If index $l$ or token $x^l$ is described as masked, then $x^l = \mathtt{m}$ and if unmasked, $x^l \neq \mathtt{m}$. Finally, for a random variable $X = (X^1, X^2, \ldots, X^d)$ and a set $A \subseteq \mathcal{I}$, we define the notation for conditional probabilities $p(x^l|x^A) = \mathbb{P}(X^l = x^l|\{X^i = x^i|i \in A\})$.

### 2.2 Masked diffusion models

Given a finite set of samples $\mathcal{D} \subseteq \mathcal{S}$ drawn from an unknown target data distribution $X \sim q$, most MDMs, with few exceptions, have been proven to learn to model clean data conditionals $q(x^l|x^{\bar{M}})$ (Ou et al., 2025; Zheng et al., 2024) where $\bar{M} \subseteq \mathcal{I}$ is a set, possibly empty, of unmasked token indices. The main difference from BERT-like masked language modeling (Devlin et al., 2019) is that BERT-like models are trained on a fixed masking ratio, as opposed to all possible masks. In particular, the MDMs we consider learn factorized conditional predictions of $p^\theta(x^l|x^{\bar{M}})$, using full attention, for all $l$ masked tokens, that ideally match $q(x^l|x^{\bar{M}})$, for every possible $\bar{M}$. Sampling

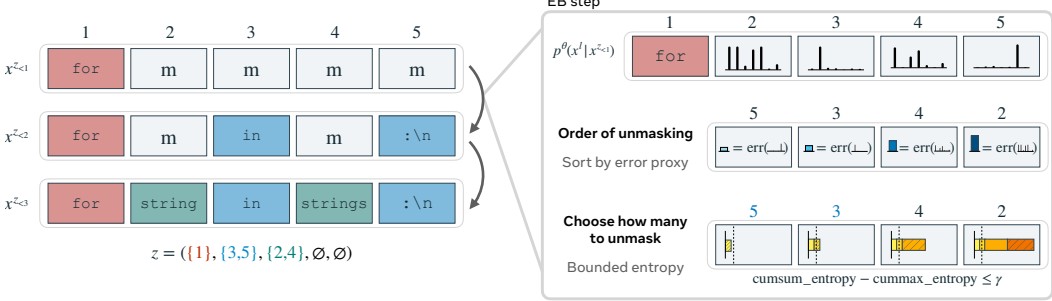

Figure 1: Illustration of an unmasking step with EB-Sampler. At each step EB sampler determines which tokens to unmask by ordering according to an error proxy and then chooses how many tokens to independently unmask by bounding their joint dependence via model prediction entropies.

with trained factorized conditionals can be done sequentially, unmasking one token at a time. This baseline sampler can be defined as follows: At each step (i) randomly pick an index to unmask, *e.g.*, $l$; (ii) sample from learned factorized conditional distribution $X^l \sim p^\theta(x^l|x^{\bar{M}})$, where $\bar{M}$ is the set of currently masked tokens. We refer to this sampling procedure as *random unmasking*.

## 3 Known challenges of MDM sampling

We now discuss two challenges in sampling from MDMs. The connections we draw between these challenges will support the intuition for the construction of the EB-Sampler in the next section.

### 3.1 Order of unmasking matters

For the optimal factorized conditionals predictor, the order of sequential unmasking will not change the underlying model distribution, $p^\theta(x)$. That is, by the chain rule of probability, for any two permutations (orders of unmasking) $\sigma, \sigma'$:

$$p_\sigma^\theta(x) = p_{\sigma'}^\theta(x)$$

where $p_\sigma^\theta = \prod_{l=1}^d p^\theta(x^{\sigma(l)}|x^{\cup_{j<l}\sigma(j)})$. Nevertheless, training an optimal $p^\theta$ is intractable due to both learning on extremely high dimensional state spaces with limited model capacity and finite data and no hard constraints on the modeling itself such that the chain rule holds. Formally, it means that there exists *local model error* $D_{\mathrm{KL}}(q(x^l|x^{\bar{M}}), p^\theta(x^l|x^{\bar{M}})) > 0$. A key consequence is that the *total model error*, *i.e.*, the discrepancy between the data distribution, $q(x)$, and the model distribution for a certain order of unmasking,

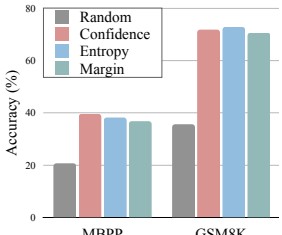

Figure 2: Performance of greedy sampling with various unmasking criteria from LLaDa 8B Base model.

$p_\sigma^\theta(x)$, depends on the *order of unmasking*. A question that arises then is, do MDMs manage to learn such there are unmasking orders with low total model error? If so, can one find local model error proxies such that the chosen unmasking order will result in low total model error?

Recent works (Nie et al., 2025a,b; Kim et al., 2025; Ye et al., 2025) proposed sequential greedy sampling with unmasking orders dictated by one of the following criteria (Chang et al., 2022; Kim et al., 2025; Zheng et al., 2023) for choosing the next coordinate $l$ to unmask:

$$\text{Confidence: } l = \underset{l' \in M}{\arg\max} \left[ \max_{x^{l'}} p^\theta(x^{l'}|x^{\bar{M}}) \right]$$

$$\text{Entropy: } l = \underset{l' \in M}{\arg\min} \left[ H(X^{l'}|X^{\bar{M}} = x^{\bar{M}}) \right] \tag{1}$$

$$\text{Margin: } l = \underset{l' \in M}{\arg\max} \left[ p^\theta(X^{l'} = y_1|x^{\bar{M}}) - p^\theta(X^{l'} = y_2|x^{\bar{M}}) \right]$$

where $(y_1, y_2) = \arg \underset{x^{l'}}{\mathrm{Top}_2}\, p^\theta(x^{l'}|x^{\bar{M}})$, where $\arg \mathrm{Top}_2$ returns the two token values with maximal probability, and $H(p) = -\sum_x p(x)\log p(x)$ is the entropy of $p$. Greedy samplers show superior performance compared to random unmasking samplers, as shown in Figure 2, closing the gap in performance between MDMs and ARMs (Nie et al., 2025b; Ye et al., 2025). This answers the first part of the question above, indicating the existence of orders with lower total model error. As for the second part, on how to find those unmasking orders, Figure 2 shows that the criteria in Equation (1) can serve as local model error proxies, determining an unmasking order with low(er) total model error.

## 3.2 Sampling efficiency

The best performing sampling procedures for language MDMs described above, similarly to ARMs, predict one token per function evaluation, hence the efficiency of MDMs remains a disadvantage compared to ARMs due to costly computations of full attention that does not allow KV-caching. An avenue for improving sampling efficiency of MDMs is to make use of the model predictions on all masked tokens to unmask multiple tokens per function evaluation. Common multi-token unmasking procedures unmask a fixed number of tokens, $k$, at each step by sampling independently from the predicted factorized conditionals. We refer to these approaches as Top-$k$ sampling, and they can all be cast in terms of choosing the Top-$k$ lowest model error proxy tokens to be unmasked at each step.

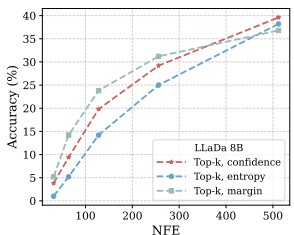

Figure 3: Efficiency-accuracy tradeoff of Top-$k$ (NFE) sampling on MBPP.

The parameter $k$ tunes an efficiency-accuracy tradeoff. The larger $k$ is, the larger the *joint dependence error* will be, as it wrongly assumes independence of a fixed number of tokens at every step. Figure 3 empirically shows the degradation in performance in Top-$k$ sampling for $k \in \{1, 2, 4, 8, 16\}$ with various error proxies.

## 4 Entropy Bounded (EB) Sampler

The previous section described how challenges in MDM sampling come from two distinct sources of error: local model error and joint dependence error. Controlling these errors motivates our *Entropy Bounded (EB) Sampler*, a direct replacement for Top-$k$ samplers.

Section 3.1 found that not only do masked tokens with low model error likely exist, but that past research has already identified proxies computable via model predictions that identify these tokens in Equation (1). A step in EB-Sampler begins by sorting unmasked tokens in ascending order on this error proxy, exactly as in Top-$k$ samplers. Then Section 3.2 showed Top-$k$ sampling accumulates substantial joint dependence error that harms performance by sampling tokens independently. We now make the additional observation that MDM predictions are often highly confident about multiple masked tokens simultaneously. These tokens are predicted to have low dependence in the data distribution $q$, because these tokens are all predicted to have low entropy. EB-Sampler therefore defines a threshold $\gamma \geq 0$ and decides to unmask the largest subset $U$ of sorted masked tokens such that

$$\sum_{l \in U} H(p^\theta(x^l|x^{\bar{M}})) - \max_{l \in U} H(p^\theta(x^l|x^{\bar{M}})) \leq \gamma. \tag{2}$$

When $U$ is comprised of low model error tokens, this expression approximately bounds a rigorous joint dependence error introduced later in Section 5. Figure 1 visually illustrates a step of EB-Sampler.

Python code showing Top-$k$ sampling and EB-Sampler is provided in Figure 4, where EB-Sampler is shown to be a minimal change in PyTorch. Like Top-$k$ sampling, EB-Sampler is easily compatible with ad hoc adjustments like temperature and unsupervised classifier-free guidance that alter $p^\theta$.

Consider this code with the same inputs. Different outputs are only from EB-Sampler determining the value of $k$ using the entropy bound of Equation (2). When unmasked tokens have low dependence, EB-Sampler will unmask more tokens, and conversely, when there is (potentially) high dependence, EB-Sampler will unmask less tokens. The amount of tokens unmasked per step in EB-Sampler is therefore not fixed. More function evaluations are used when the sample is predicted complex and less when the sample is predicted simple. The threshold $\gamma$ influences the number of steps where $\gamma = 0$ will unmask one token each step and $\gamma = \infty$ will unmask all tokens at once.

```
1   def top_k_step(x, model, sample_fn, error_proxy_fn, k):      def EB_step(x, model, sample_fn, error_proxy_fn, gamma):
2       p = model(x)                                                 p = model(x)
3       err = error_proxy_fn(p)                                      err = error_proxy_fn(p)
4       err = torch.where(x == model.mask_id, err, np.inf)           err = torch.where(x == model.mask_id, err, np.inf)
5       _ , ids = torch.sort(err, dim=-1)                            _ , ids = torch.sort(err, dim=-1)
6                                                                 +  entropy = torch.distributions.Categorical(probs=p).entropy()[ids]
7                                                                 +  acc_entropy = torch.cumsum(entropy)
8                                                                 +  cummax_entropy = torch.cummax(entropy, dim=0).values
9                                                                 +  k = (acc_entropy - cummax_entropy <= gamma).sum()
10      k = torch.minimum(k, (x == model.mask_id).sum())             k = torch.minimum(k, (x == model.mask_id).sum())
11      ids_to_unmask = ids[:k]                                      ids_to_unmask = ids[:k]
12      x[ids_to_unmask] = sample_fn(p, ids_to_unmask)               x[ids_to_unmask] = sample_fn(p, ids_to_unmask)
13      return x                                                     return x
```

Figure 4: Python code implementation of a single sampling step for common Top-$k$ approaches and for EB-Sampler.

## 5   Adaptive unmasking samplers

In this section, we formulate the EB-Sampler as a member of a more general family of adaptive multi-token samplers. The object we use to mathematically describe varying length unmasking steps is an ordered partition $\mathcal{I}$, denoted by $z = (z_1, z_2, \ldots, z_d)$, where $z$ satisfies:

$$\bigcup_{i=1}^{d} z_i = \mathcal{I}, \quad z_i \cap z_j = \varnothing \tag{3}$$

and either $z_i \subseteq \mathcal{I}$ or $z_i = \varnothing$. Notation $z_{<i}$ denotes ordered sub-partitions up to index $i$, that is $z_{<i} = (z_j | j \in [i-1])$.

For a random variable $X = (X^1, X^2, \ldots, X^d)$ over $\mathcal{S}$ and ordered sub-partitions $s, s'$, we extend the notation from Section 2.1 for conditional probabilities:

$$p(x^s | x^{s'}) = \mathbb{P}\left(\left\{X^l = x^l, \ \forall l \in \mathcal{I}_s\right\} \ \middle| \ \left\{X^j = x^j, \ \forall j \in \mathcal{I}_{s'}\right\}\right) \tag{4}$$

where $\mathcal{I}_s = \bigcup_{i=1}^{|s|} s_i$ and $\mathcal{I}_{s'} = \bigcup_{i=1}^{|s'|} s'_i$. Depending on context, we will also be using the notation in 4 with $s$ being a set of indices, $e.g.$, $s = z_i$ or a single index, $e.g.$, $s = l$.

We consider a broad family of sampling procedures defined by $\phi$ that leverage approximate clean data conditionals provided by $p^\theta$, and produce a joint distribution $p_\phi(x, z)$ over state $x$ and partition $z$:

$$p_\phi(x, z) = \prod_{i=1}^{d} p_\phi(z_i, x^{z_i} | x^{z_{<i}}, z_{<i}) = \prod_{i=1}^{d} \left(\prod_{l \in z_i} p^\theta(x^l | x^{z_{<i}})\right) \phi(z_i | x^{z_{<i}}, z_{<i}). \tag{5}$$

The distributions $\phi$ enforce $z$ is a valid partition, and sampling $z_i$ means token indices $z_i$ are unmasked at step $i$. Without loss of generality, $\phi$ always unmasks at least one token if possible, i.e. only samples $z_i = \varnothing$ when $z_{<i} = \mathcal{I}$. Similarly define

$$q_\phi(x, z) = \prod_{i=1}^{d} q(z_i, x^{z_i} | x^{z_{<i}}, z_{<i}) = \prod_{i=1}^{d} q(x^{z_i} | x^{z_{<i}}) \phi(z_i | x^{z_{<i}}, z_{<i}). \tag{6}$$

Importantly, $q_\phi(x, z) = q(x) \prod_{i=1}^{d} \phi(z_i | x^{z_{<i}}, z_{<i})$, because the product of the clean data conditionals does not depend on the order of unmasking, $z$, so that $\sum_z q_\phi(x, z) = q(x)$ for any $\phi$.

**Expressiveness of** $\phi$   This family encompasses existing common samplers that always unmask the same number of tokens on each step, such as deterministic samplers of Top-$k$ smallest margin, Top-$k$ smallest entropy, and Top-$k$ confidence, or simple random unmasking, but also contains additional options, including dynamically determining the number of tokens to unmask at each step.

**Error decomposition**   We quantify the error from sampling $p_\phi(x) = \sum_z p_\phi(x, z)$ instead of $q(x)$ via KL divergence

$$D_{\mathrm{KL}}(q(x), p_\phi(x)) \leq D_{\mathrm{KL}}(q_\phi(x, z), p_\phi(x, z)) = \sum_{i=1}^{d} \mathbb{E}_{q_\phi}[\ln q(x^{z_i} | x^{z_{<i}}) - \sum_{l \in z^i} \ln p^\theta(x^l | x^{z_{<i}})].$$

Appendix A.1 discusses when this is an equality, and proves this can be rewritten into two terms

$$\sum_{i=1}^{d} \mathbb{E}_{q_\phi} [\underbrace{\sum_{l \in z_i} D_{\mathrm{KL}}(q(x^l|x^{z<i}), p^\theta(x^l|x^{z<i}))}_{\text{model error}} + \underbrace{D_{\mathrm{KL}}(q(x^{z_i}|x^{z<i}), \prod_{l \in z_i} q(x^l|x^{z<i}))}_{\text{joint dependence error}}]. \qquad (7)$$

*Model error* comes from sampling incorrect conditionals $p^\theta$ that do not match the data distribution. *Joint dependence error* comes from sampling tokens independently that are not actually independent in $q$. This second KL divergence is precisely joint mutual information, upper-bounded by

$$D_{\mathrm{KL}}(q(x^{z_i}|x^{z<i}), \prod_{l \in z_i} q(x^l|x^{z<i})) \leq \sum_{l \in z_i} H(q(x^l|x^{z<i})) - \max_{l \in z_i} H(q(x^l|x^{z<i})). \qquad (8)$$

**Choosing $\phi$ given a pre-trained model** EB-Sampler is directly motivated by this error decomposition. The $p^\theta$ that achieves zero model error is the same for any $\phi$, justifying using any pre-trained model learned to match clean data conditionals. Assume we can identify low model error tokens and we design $\phi$ to only select $z_i$ from such tokens, where for all $l \in z_i$, $p^\theta(x^l|x^{z<i}) \approx q(x^l|x^{z<i})$. Then model error is negligible and joint dependence error is approximately upper-bounded by

$$\sum_{l \in z_i} H(p^\theta(x^l|x^{z<i})) - \max_{l \in z_i} H(p^\theta(x^l|x^{z<i})), \qquad (9)$$

our criteria in Equation (2). So after adaptively identifying low model error tokens, $\phi$ can control overall error by selecting subsets of such tokens to unmask with bounded joint dependence. EB-Sampler applies this approach with the model error proxies from Section 3.1.

## 6 Experiments

We evaluate the performance of the EB-Sampler on standard code and math reasoning generation tasks and on logic puzzles solving. The empirical findings in this section support our theoretical derivations and demonstrate the proposed sampler's capabilities.

**Baselines.** We compare the EB-Sampler to Top-$k$ samplers with the three error proxy functionals described in 1: (i) confidence; (ii) entropy; and (iii) margin.

**Experimental setting.** On all tasks and models we follow the same general experimental setting. We test the EB-Sampler with a range of thresholds, $\gamma$, depending on the task. For the Top-$k$ samplers we test a range of $k$ values. Each point in the plots (*e.g.*, Figure 5) corresponds to the resulting model performance when sampling with parameter $\gamma$ or $k$ for EB or Top-$k$ sampler respectively. After finding temperature was unhelpful for LLaDa, we used zero temperature sampling for all experiments. Full details on threshold and $k$ values can be found in Appendix C.

### 6.1 Code and math reasoning

We evaluate the performance of the EB-Sampler variants on text generation tasks in which (i) success can be measured quantitatively; and (ii) require an answer that is longer than a single token. These two properties facilitate quantitative assessment of the efficiency-accuracy tradeoff a sampler exhibits.

**Models.** We report results on two recent open source state-of-the-art language MDMs: LLaDa 8B Base (Nie et al., 2025b) and Dream 7B Base (Ye et al., 2025).

**Benchmarks.** We use 4 widely used benchmarks. HumanEval (0 shot) (Chen et al., 2021) and MBPP (4 shot) (Austin et al., 2021b) code generation benchmarks; and GSM8K (8 shot) (Cobbe et al., 2021) and Math (4 shot) (Hendrycks et al., 2021) math reasoning benchmarks. We note that we use different variants of GSM8K and Math in our evaluation compared to reported results for Dream 7B, hence the minor gaps in performance for standard Top-1 sampling.

**Setup.** For each task, we follow common practice and set a maximal number of generated tokens, `max_gen_len`. Then, the prompt is padded to the right with mask tokens until the maximal sequence length of the model. During sampling, only masked tokens in the range of $[\text{len(prompt)}, \text{len(prompt)} + \text{max\_gen\_len}]$ are allowed to be unmasked. Generation stops once all tokens in the designated range have been unmasked. In the main text, we show results for the two best performing error proxy functions entropy and confidence. Results with margin are in Appendix D.

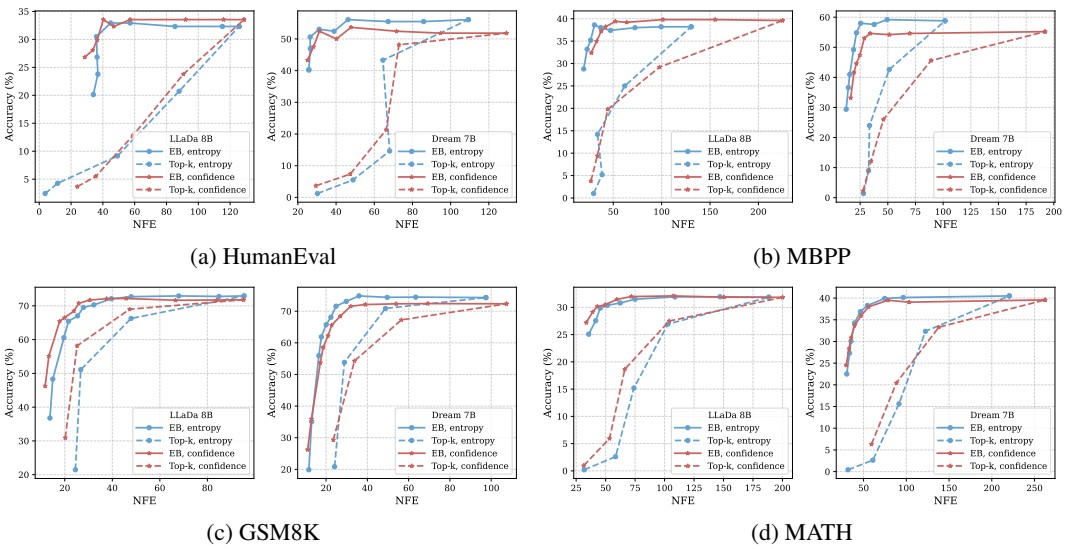

(a) HumanEval  (b) MBPP

(c) GSM8K  (d) MATH

Figure 5: pass@1 accuracy vs. NFE with `generate_until` logic on code and math reasoning tasks.

### 6.1.1 Measuring efficiency gains of MDM samplers

In this part, we describe how we measure the gains obtained by our proposed sampler. We explain why current generation practices for MDMs require rethinking and propose to add a `generate_until` logic to save function evaluations. We then further observe that unlike ARMs that generate in a left-to-right order, a `generate_until` logic may still be suboptimal for MDM efficient generation.

**`generate_until` logic.** Current generation practices for MDMs fix an apriori amount of tokens to be generated, *i.e.*, `max_gen_len`, and generate until all tokens are unmasked. This is rather wasteful as typically producing a response to a given prompt will require less tokens than the full length, that is `answer_len < max_gen_len`. We therefore propose to incorporate a `generate_until` logic in MDM samplers similar to ARM generation practices. For example, for the MBPP benchmark, the few shot samples given as context to the model include a concluding phrase "[DONE]". In ARM evaluation procedures, this phrase is used as a stopping criterion. For MDMs, however, due to the non left-to-right order of unmasking, we extend the `generate_until` logic to include an additional condition that all tokens in indices preceding the concluding phrase are unmasked.

All experiments were run with the `generate_until` logic applied as a post-process in order to measure gains both against full `max_gen_len` and effective generation length to stopping criterion. The NFE reported in Figure 5 measures average number of function evaluations until a task specific `generate_until` logic is satisfied. EB-Sampler consistently improves upon accuracy-NFE Pareto frontier across all datasets and error proxies, gaining speed-ups of 2-4x compared to Top-1 sampler at the same accuracy. In Figures 9 and 10 we show the same plots against full `max_gen_len` NFE.

**Bias in function evaluation count.** Surprisingly, under the `generate_until` post-process we still observed high NFE counts at $\gamma = 0$ compared to expected answer lengths. This is most pronounced on the MBPP benchmark, where the model finds it "easy" to repeat the instructions given to the model after the concluding phrase although it did not finish unmasking all the masks before that. To get a better estimate of the actual speed-up gains achieved by the EB-Sampler we take MBPP as a test case and adopt the semi-AR block generation scheme from (Arriola et al., 2025; Nie et al., 2025b) to restrict the model from generating tokens that are far from the context. In Table 1 we compare the average NFE at roughly the same performance for the various strategies of controlling the generation length. We note that for this dataset and the sampling strategies evaluated in the table the mean answer length is around 50 tokens, while the semi-AR block generation requires 64.59 NFE, thus not fully resolving the bias. On the contrary, EB-Sampler requires 21.19 function evaluations to get the same performance with same average answer length, generating tokens at a rate of 2.4 tokens per step. We therefore refrain from claiming a 6x speed up as it compares to a loose generation procedure and believe a better estimate of the EB-Sampler's gains is around 2-3x. Table 1 provides three key insights on MDM efficiency evaluation. First, measuring the efficiency gains of samplers for MDMs

Table 1: NFE and Speed-Ups for Dream 7B on the MBPP for various evaluation schemes at roughly same best pass@1. For all configurations in the table the mean answer length is $\sim 50$ tokens.

| | Full `max_gen_len = 512` | | | `generate_until` logic | | `generate_until` logic + semi-AR (block_len=64) | | |
|---|---|---|---|---|---|---|---|---|
| | pass@1 | NFE | Speed-Up | NFE | Speed-Up | pass@1 | NFE | Speed-Up |
| Top-1 | 58.8% | 512 | x 1 | 101.71 | x 1 | 58.8% | 64.59 | x 1 |
| EB, entropy, $\gamma = 0.001$ | 59.2% | 174.57 | x 2.93 | 49.39 | x 2.05 | 59% | 38.90 | x 1.66 |
| EB, entropy, $\gamma = 0.1$ | 58% | 85.33 | x 6.00 | 25.49 | x 3.99 | 58.6% | 21.19 | x 3.05 |

is non-trivial and depends on the generation configuration, second, MDMs invest computation on generating tokens that are not used and this opens up an important practical question for the future use of MDMs, and lastly, EB-Sampler shows strong performance in efficiency gains across all settings.

## 6.2 Logic puzzles

Discrete diffusion models have been shown to excel on logic puzzles such as maze navigation, Sudoku and more (Nolte et al., 2024; Ye et al., 2024). We investigate whether these strong performances can be retained while sampling more efficiently than one token at a time. Specifically, we train small scale discrete diffusion models on Sudoku and Maze navigation problems and evaluate their performance on held-out data when varying the sampling strategies.

### 6.2.1 Maze navigation

We use the maze generation methods for "DFS mazes" in (Nolte et al., 2024), see their Section 4.1. We generate $48K$ mazes for training and $2K$ for validation. All mazes are defined on a grid of size 10x10. They are serialized into tokens by enumerating all the connections between cells in the grid, i.e. the edges in the graph defined by the maze. To aid learning the invariance with respect to edge reordering, the edges are shuffled before tokenization. A 6 million parameter discrete DiT model, using code adapted from (Lou et al., 2023), is trained to optimize the masked diffusion objective, without explicit time dependence (Kitouni et al., 2024; Ou et al., 2025).

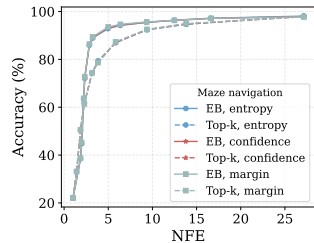

Figure 6: 10x10 mazes - accuracy vs average NFE.

The metric used for performance comparison is the accuracy, defined as the fraction of validation mazes fully solved. Figure 6 shows the accuracy as a function of average NFE over the validation set for the different ordering metrics. All metrics perform extremely similarly, thus partially obscured in the plot. All strategies work best in the single token unmasking regime. EB-Sampler preserves most of the accuracy before experiencing a sharp drop-off at less than 5 NFEs. In contrast, the Top-$k$ baselines exhibit a steeper decline in performance, already at around 10 NFEs.

### 6.2.2 Sudoku

To further assess sampling strategies in structured logic problems, we evaluate performance on the task of Sudoku completion. Unlike maze navigation, which emphasizes path finding, Sudoku requires reasoning over dense, globally constrained grids. This setting provides a complementary benchmark to test sampling strategies under those global constraints.

We adopt the standard 9×9 Sudoku setting and adapt the code from Alp (2024) to generate $48K$ training puzzles and $2K$ held-out puzzles with, all with unique solutions. Each puzzle is serialized into a sequence of 89 tokens corresponding to the cell values and end-of-line tokens, with zeros indicating blank cells. The discrete DiT model architecture used for maze navigation is reused here.

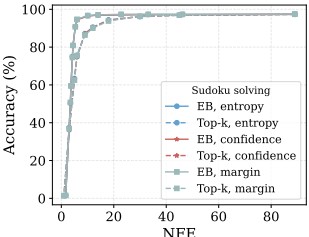

Figure 7: Sudoku - accuracy vs average NFE.

We again show accuracy as a function of average NFE. The results are depicted in Figure 7. The trend is similar to the trend in the maze navigation setup. EB-Sampler retains most of its performance for a longer time than the Top-$k$ samplers, and then drops off sharply. Notably, almost full performance is retained even when averaging around 10-15 NFEs.

# 7 Related Work

**Performant sampling for discrete diffusion.** Procedures that improve sampling from MDMs are often focused on improving performance for a given pre-trained model. Recent approaches consider *planning* (Kim et al., 2025), deciding which masked tokens should be unmasked next, as well as *remasking* (Wang et al., 2025), deciding which unmasked tokens should be masked again, related to predictor-corrector iterations (Gat et al., 2024; Lezama et al., 2022) and forward-backwards sampling (Campbell et al., 2024), or consider both planning and remasking in (Zheng et al., 2023; Peng et al., 2025; Liu et al., 2024). Like EB-Sampler, this research often considers KL (equivalently ELBO) bounds. Unlike past research though, we focus on multi-token adaptive planning with a variable-sized set of tokens to unmask, crucial for efficiency. While sampling for LLaDa (Nie et al., 2025b) was described as *remasking*, it can be viewed as determining what to unmask first and then their token values within semi-autoregressive blocks, a member of our $\phi$ family. Because we aim for efficient and justified planning for scaled MDMs, we do not consider revisiting unmasked tokens here, enabling a simpler, time-independent analysis with a KL bound that is minimized via adaptive sampling. Future research might devise an efficient multi-token sampler that both unmasks and revisits past tokens upon EB-Sampler. One option might be to combine EB-Sampler with predictor-corrector iterations suggested in (Zhao et al., 2024) that update unmasked variables but do not change masks.

**Efficient sampling for discrete diffusion.** Efficiency has received relatively less attention than performance. (Ren et al., 2025) proposed higher-order numerical solvers for discrete diffusion, not specific to MDMs. (Park et al., 2024) introduced a method to avoid joint dependence error from parallel sampling, by performing a global optimization for a non-adaptive sampling schedule (i.e. the number of tokens per step). EB-Sampler determines this adaptively per sampling step. (Besnier et al., 2025) introduced an unmasking sampler for MaskGIT (Chang et al., 2022) that controls joint dependence error via quasi-random, low-discrepancy unmasking of an image, outperforming a confidence-based sampler in that domain. Finally, recent research (Zhu et al., 2025) has proposed a distillation procedure for MDMs, and trained one-step image generators from multi-step MaskGIT (Chang et al., 2022) and Messionic (Bai et al., 2024). Other works, analyzing the convergence and error of discrete diffusion models such as Chen and Ying (2024); Liang et al. (2025); Zhang et al. (2025) could potentially enable extension of our analysis of time-independent masked model to time dependent discrete diffusion.

**Speculative decoding.** A prominent method to accelerate LLMs is speculative decoding, with 2-2.5x speedup in (Chen et al., 2023). Instead of sampling from a large *target* language model, speculative decoding generates a candidate sequence from a smaller *draft* language model and accepts some portion of that candidate utilizing sequence probabilities. Modified rejection sampling guarantees the sequence is extended and this extension is sampled from the target model. Because evaluating models with causal attention can be done cheaply compared to sampling, efficiency gains occur when the draft model quickly samples reasonable sequences. This procedure has also been adapted to any-order masked models (Uria et al., 2014; Hoogeboom et al., 2022) with causal attention (Pannatier et al., 2024; Guo and Ermon, 2025). For MDMs with full attention, we cannot directly apply speculative decoding because it is generally expensive to compute the sequence probability in a full attention target model. (De Bortoli et al., 2025) has applied speculative sampling to continuous Gaussian diffusion, but relies upon querying the target model in parallel. Speculative decoding has been combined with discrete diffusion in (Christopher et al., 2024), where the draft is an MDM and the target is a language model. Our EB-Sampler approach is complementary, and could be simply applied to speed up the draft model.

# 8 Conclusions and Future Work

In this paper we propose EB-Sampler, a theoretically grounded adaptive sampler for masked discrete diffusion and flow models. This algorithm controls both which and how many tokens to sample using an interpretable entropy bound and serves as a drop-in replacement for existing samplers. We evaluate our approach on math, code and reasoning benchmarks with contemporary diffusion models of different scales, against the standard samplers used by the authors of those models. We find that EB-Sampler significantly outperforms existing samplers on the compute-vs-performance Pareto frontier and even yields 2-3x speed-ups without any loss of performance.

Future research might consider learning a parameterized adaptive sampler from data, perhaps optimizing our KL bound with respect to $\phi$, or expand upon EB-Sampler to incorporate revisiting past unmasked tokens. Such research could further advance efficient and performant sampling for masked diffusion models.

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

# A  Theorems and proofs

## A.1  KL divergence error decomposition

We revisit the KL divergence introduced in Section 5. Recall we quantify the error from sampling $p_\phi(x) = \sum_z p_\phi(x, z)$ instead of $q(x)$ via KL divergence

$$D_{\mathrm{KL}}(q(x), p_\phi(x)) \le D_{\mathrm{KL}}(q_\phi(x, z), p_\phi(x, z)) = \sum_{i=1}^d \mathbb{E}_{q_\phi}\left[\ln q(x^{z_i}|x^{z<i}) - \sum_{l \in z^i} \ln p^\theta(x^l|x^{z<i})\right],$$
(10)

where the inequality can be derived from the Evidence Lower BOund (ELBO) (Kingma and Welling, 2019). In the equality we plug in the definitions of the joint distributions $p_\phi(x, z), q_\phi(x, z)$ from Equation (5) and Equation (6) respectively, where the $\phi$ term cancels out, and the sum and expectation can be interchanged since the sum does not

We add and subtract $\ln\left(\prod_{i=1}^d \prod_{l \in z^i} q(x^l|x^{z<i})\right)$ from Equation (10):

$$\sum_{i=1}^d \mathbb{E}_{q_\phi}\left[\ln q(x^{z_i}|x^{z<i}) - \sum_{l \in z^i} \ln p^\theta(x^l|x^{z<i})\right]$$
$$= \sum_{i=1}^d \mathbb{E}_{q_\phi}\left[\sum_{l \in z^i} \ln \frac{q(x^l|x^{z<i})}{p^\theta(x^l|x^{z<i})} + \ln \frac{q(x^{z_i}|x^{z<i})}{\prod_{l \in z^i} q(x^l|x^{z<i})}\right],$$
(11)

and will now separately simplify the two terms.

**Joint dependence error.** We begin with the right term of the expectation in last equality of Equation (11), which will turn out to be the joint dependence error term in Equation (7).

$$\mathbb{E}_{q_\phi}\left[\ln \frac{q(x^{z_i}|x^{z<i})}{\prod_{l \in z^i} q(x^l|x^{z<i})}\right] =$$
$$= \sum_{x,z} q_\phi(x, z)\left(\ln \frac{q(x^{z_i}|x^{z<i})}{\prod_{l \in z^i} q(x^l|x^{z<i})}\right) =$$
$$\overset{(*)}{=} \sum_{z_{\le i}, x^{z \le i}} q_\phi(z_{\le i}, x^{z \le i})\left(\ln \frac{q(x^{z_i}|x^{z<i})}{\prod_{l \in z^i} q(x^l|x^{z<i})}\right) =$$
$$= \sum_{z_{\le i}, x^{z<i}} q_\phi(z_{\le i}, x^{z<i}) \sum_{x^{z_i}} q_\phi(x^{z_i}|x^{z<i}, z_{\le i})\left(\ln \frac{q(x^{z_i}|x^{z<i})}{\prod_{l \in z^i} q(x^l|x^{z<i})}\right) =$$
$$\overset{(**)}{=} \mathbb{E}_{q_\phi(z_{\le i}, x^{z<i})}\left[\sum_{x^{z_i}} q(x^{z_i}|x^{z<i}) \ln \frac{q(x^{z_i}|x^{z<i})}{\prod_{l \in z^i} q(x^l|x^{z<i})}\right] =$$
$$= \mathbb{E}_{q_\phi(z_{\le i}, x^{z<i})}\left[D_{\mathrm{KL}}\left(q(x^{z_i}|x^{z<i}), \prod_{l \in z^i} q(x^l|x^{z<i}))\right)\right].$$
(12)

where in $(*)$ we marginalize over $x^{z>i}$ and $z_{>i}$ since the function the expectation is taken over does not depend on them. In $(**)$ we use the fact that for the sampling procedure defined by $\phi$, $q_\phi(x^{z_i}|x^{z<i, z \le i}) = q(x^{z_i}|x^{z<i})$. At last, we arrive at an expectation of the KL-divergence between the joint and the factorized product distributions of $x^{z_i}$ conditioned on all the unmasked tokens before, measuring the joint dependence withoun the subset of indices $z_i$.

**Model error.** As for the left term in the last equality of Equation (11), it will turn out to be the model error term in Equation (7). Recalling the left term:

$$
\mathbb{E}_{q_\phi} \left[ \sum_{l \in z^i} \ln \frac{q(x^l | x^{z<i})}{p^\theta(x^l | x^{z<i})} \right] =
$$

$$
= \sum_{x,z} q_\phi(x,z) \left( \sum_{l \in z^i} \ln \frac{q(x^l | x^{z<i})}{p^\theta(x^l | x^{z<i})} \right) =
$$

$$
\stackrel{(*)}{=} \sum_{z_{\leq i}, x^{z \leq i}} q_\phi(z_{\leq i}, x^{z \leq i}) \left( \sum_{l \in z^i} \ln \frac{q(x^l | x^{z<i})}{p^\theta(x^l | x^{z<i})} \right) =
$$

$$
= \sum_{z_{\leq i}, x^{z<i}} q_\phi(z_{\leq i}, x^{z<i}) \sum_{x^{z_i}} q_\phi(x^{z_i} | x^{z<i}, z_{\leq i}) \left( \sum_{l \in z^i} \ln \frac{q(x^l | x^{z<i})}{p^\theta(x^l | x^{z<i})} \right) =
$$

$$
\stackrel{(**)}{=} \mathbb{E}_{q_\phi(z_{\leq i}, x^{z<i})} \left[ \sum_{l \in z^i} \sum_{x^{z_i}} q(x^{z_i} | x^{z<i}) \ln \frac{q(x^l | x^{z<i})}{p^\theta(x^l | x^{z<i})} \right] =
$$

$$
\stackrel{(***)}{=} \mathbb{E}_{q_\phi(z_{\leq i}, x^{z<i})} \left[ \sum_{l \in z^i} \sum_{x^l} q(x^l | x^{z<i}) \ln \frac{q(x^l | x^{z<i})}{p^\theta(x^l | x^{z<i})} \right] =
$$

$$
= \mathbb{E}_{q_\phi(z_{\leq i}, x^{z<i})} \left[ \sum_{l \in z^i} D_{\mathrm{KL}}(q(x^l | x^{z<i}), p^\theta(x^l | x^{z<i})) \right]. \tag{13}
$$

where $(*)$ and $(**)$ are the same steps as in the joint dependence error derivation above. In $(***)$, we again marginalize over variable that do not appear in the expectation. At last, we arrive at a sum of of the KL-divergences between the factorized conditionals, which is exactly what $p^\theta$ is trained to learn, and this we call this term *model error*.

**KL divergence equality:** The two KL divergences, $D_{\mathrm{KL}}(q(x), p_\phi(x))$ and $D_{\mathrm{KL}}(q_\phi(x,z), p_\phi(x,z))$, are equal when $q_\phi(z|x) = p_\phi(z|x)$. This only occurs under special $\phi$. For any $\phi$,

$$
q_\phi(z|x) = \frac{q_\phi(x,z)}{q(x)} = \frac{q(x) \prod_{i=1}^d \phi(z^i | x^{z<i}, z^{<i})}{q(x)} = \prod_{i=1}^d \phi(z^i | x^{z<i}, z^{<i}). \tag{14}
$$

Then

$$
p_\phi(z|x) = \frac{p_\phi(x,z)}{p_\phi(x)}
$$

$$
= \frac{\prod_{i=1}^d \left( \prod_{l \in z^i} p^\theta(x^l | x^{z<i}) \right) \phi(z^i | x^{z<i}, z^{<i})}{p_\phi(x)}
$$

$$
= \frac{\prod_{i=1}^d \left( \prod_{l \in z^i} p^\theta(x^l | x^{z<i}) \right)}{p_\phi(x)} q_\phi(z|x). \tag{15}
$$

Thus $p_\phi(z|x)$ is almost certainly not equal to $q_\phi(z|x)$ even if we unmask one token sequentially, because $p^\theta$ is learned. The product of the learned conditionals almost certainly results in a different joint distribution depending on the order, unlike for the data distribution where the product of the true conditionals is $q(x)$ for every order.

However, when every $\phi$ is deterministic we do have equality. This special case is relevant because many schemes introduced in the main text are deterministic, or nearly so. Then the partition $z$ is entirely determined by $x$ and can be written $z_\phi(x)$, and $q_\phi(z|x) = p_\phi(z|x)$ are a point mass on $z_\phi(x)$. For a deterministic $\phi$ optimizing the KL divergence hence directly optimizes the likelihood $\mathbb{E}_q[\ln p_\phi(x)]$, and not a lower-bound on that likelihood, because $\mathbb{E}_q[\ln p_\phi(x)] = \mathbb{E}_{q_\phi}[\ln p_\phi(x, z_\phi(x))]$ when $z$ is deterministically generated.

# B   Algorithm

---

**Algorithm 1** EB-Sampler with `generate_until` logic

---

**Require:** Factorized conditionals predictions $p^{\theta,l}(\cdot|x)$, threshold $\gamma \geq 0$, prompt $y_0$, sequence length $d$, error proxy functional $E$, entropy functional $H$, stopping criteria $C : \mathcal{S} \rightarrow \{\texttt{True}, \texttt{False}\}$, mask token $\mathtt{m}$

$n = d - \text{len}(y_0)$

$x \leftarrow [y_0, \mathtt{m} * n]$  $\qquad\qquad\qquad\qquad\qquad\qquad$ ▷ Set initial condition

$\mathcal{I}_{\mathtt{m}} = \{l | x^l = \mathtt{m}\}$

**while** $\mathcal{I}_{\mathtt{m}} \neq \emptyset$ and not $C(x)$ **do**  $\qquad\qquad\qquad\qquad$ ▷ Stop if all tokens unmasked

$\qquad \hat{p}_l = p^{\theta,l}(\cdot|x)$, for $l \in \mathcal{I}_{\mathtt{m}}$

$\qquad \hat{e} = E(\hat{p})$

$\qquad \hat{h} = H(\hat{p})$

$\qquad I_{\text{sort}} = \texttt{argsort}(\hat{e}_{\mathcal{I}_{\mathtt{m}}})$  $\qquad\qquad\qquad\qquad$ ▷ Sort masked tokens by error

$\qquad U \leftarrow \{\}$  $\qquad\qquad\qquad\qquad\qquad\qquad\qquad\qquad$ ▷ Initialize subpartition

$\qquad$ **for** $a$ in $I_{\text{sort}}$ **do**  $\qquad\qquad\qquad\qquad\qquad$ ▷ Iterate over sorted masked tokens

$\qquad\qquad U \leftarrow U \cup a$

$\qquad\qquad$ **if** $\texttt{sum}(\hat{h}_U) - \texttt{max}(\hat{h}_U) \leq \gamma$ **then**  $\qquad\quad$ ▷ Compute entropy bound equation 2

$\qquad\qquad\qquad x^a = \texttt{sample}(\hat{p}_a)$  $\qquad$ ▷ Sample unmasked value from posterior if below threshold

$\qquad\qquad$ **else**

$\qquad\qquad\qquad$ **break**  $\qquad\qquad\qquad\qquad$ ▷ Halt unmasking for loop if entropy bound is exceeded

$\qquad \mathcal{I}_{\mathtt{m}} = \{l | x^l = \mathtt{m}\}$

**return** $x$

---

# C   Experimental details

## C.1   Code and math reasoning

### C.1.1   Datasets

For code generation tasks we evaluate EB-Sampler on 0-shot HumanEval Chen et al. (2021), 4-shot MBPP Austin et al. (2021b), and for math reasoning we test 8-shot GSM8K Cobbe et al. (2021) without Chain Of Thought (COT) variant, and 4-shot Math Hendrycks et al. (2021) corresponding to the `hendrycks_math` variant.

### C.1.2   Setup

We evaluate the efficiency gains of EB-Sampler on two recent state of the art MDMs: LLaDa 8B Nie et al. (2025b) and Dream 7B Ye et al. (2025). For LLaDa 8B the maximal sequence length is 4096 and for Dream 7B, 2048. That is, the input to the models in the beginning of generation, is a padded sequence, starting with the prompt given in each task and then padded with the mask token, $\mathtt{m}$, to the maximal sequence length, denoted `max_seq_len`. For each dataset a predetermined generation length is set, denoted `max_gen_len`. We enforce unmasking tokens that are in the range $[\texttt{len(prompt)}, \texttt{len(prompt)} + \texttt{max\_gen\_len}]$. In the rare case when $\texttt{len(prompt)} + \texttt{max\_gen\_len} > \texttt{max\_seq\_len}$ the prompt is truncated from the left.

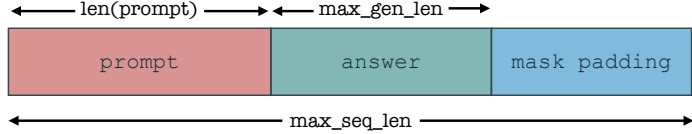

Figure 8: Input sequence visualization.

Table 2: Evaluation parameters for code and math reasoning.

| | Dataset size | #-shots | max_gen_len | generate_until phrase | |
| | | | | LLaDa 8B | Dream 7B |
|---|---|---|---|---|---|
| HumanEval | 164 | 0 | 512 | ["<\|endoftext\|>","\n\n\n"] | ["<\|endoftext\|>", ""\n"] |
| MBPP | 500 | 4 | 512 | "[DONE]" | |
| GSM8K | 1320 | 8 | 256 | "The answer is %d." | |
| MATH | 5000 | 4 | 512 | "I hope it is correct." | |

All benchmarks were run with the same $\gamma$ range of values for EB-Sampler, $\gamma \in \{0, 0.001, 0.01, 0.1, 0.25, 0.5, 0.75, 1.0, 1.5\}$, and with same $k$ range of values for Top-k sampling, $k \in \{1, 2, 4, 8\}$. All benchmarks were run in the same computational setting, on $8 \times$H100. We report runtimes for confidence and entropy error proxies in Table 3. Runtimes with margin error proxy are longer due to the need to sort over the vocabulary size to compute the top-2 tokens. Runtimes for LLaDa 8B are longer than Dream 7B due to having twice the maximal sequence length of Dream.

Table 3: Average runtimes on $8 \times$H100 of code and math benchmarks evaluation for 1 token per step sampling ($\text{Top}_1$) with entropy and confidence error proxies. Relative standard deviation of measurements is $1\%$.

| | Runtimes (hrs.) | |
| | LLaDa 8B | Dream 7B |
|---|---|---|
| HumanEval | 0.58 | 0.26 |
| MBPP | 1.75 | 0.79 |
| GSM8K | 2.29 | 1.03 |
| MATH | 17.30 | 7.84 |

### C.1.3   Post-process

**Accuracy evaluation.**    Model outputs for all datasets evaluated with both LLaDa 8B and Dream 7B had been directly fed into the standard evaluation scripts, except for HumanEval with Dream 7B. Evaluating the raw output of Dream 7B on the HumanEval benchmark results in around $8\%$ drop in performance compared to reported results by the authors. Investigating the cause for the drop in performance led to the observation that Dream 7B sometimes produces answers with a template that places the generated code inside code blocks which get ignored when compiled and are therefore considered as failure at the task. We thus post-process Dream's raw outputs on HumanEval to extract the function implementation, closing the gap to reported results to $< 2\%$. Importantly, we emphasize that all comparisons in our paper are between sampling procedures from the same model and same evaluation.

**Efficiency measurement.**    As noted in Section 6.1.1, standard sampling procedures from MDM generate a sequence with predetermined length, max_gen_len. In most cases, the answer to the given prompt will be shorter, denoted answer_len, and there are (max_gen_len − answer_len) generated tokens that are being truncated, hence unused, during evaluation. To isolate the efficiency gain EB-Sampler provides in generating the answer tokens from the gain in generating the rest of the tokens that are later not used, we incorporate the generate_until logic as a post-process. We note that this logic can also be integrated in the sampling procedure itself, saving up computation, without changing the performance of the model, as the logic ensures termination of generation only once the stopping criterion has been met. The phrases used as markers for the generate_until post-process logic for each dataset and each model are listed in Table 2.

# D   Additional experiments - code and math reasoning

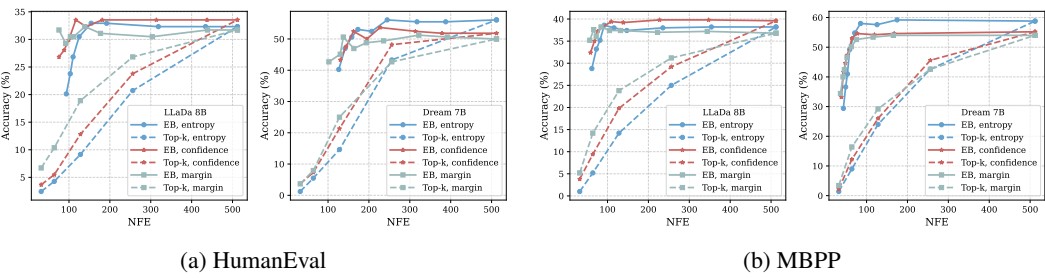

(a) HumanEval

(b) MBPP

Figure 9: pass@1 accuracy vs. full `max_gen_len` NFE on code reasoning tasks.

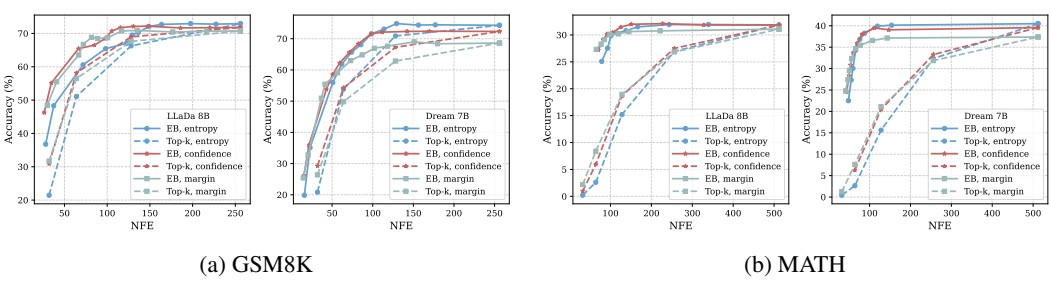

(a) GSM8K

(b) MATH

Figure 10: pass@1 accuracy vs. full `max_gen_len` NFE on math reasoning tasks.

## D.1   Results with margin error proxy

In Figure 11 we show the results with the margin error proxy along with the results with confidence and entropy error proxies presented in the main body of the paper in Figure 5. We observed that confidence error proxy was typically the best for the LLaDa 8B model and entropy error proxy worked best in most cases for Dream 7B. The margin error proxy mostly yielded inferior accuracy in full NFE, thus to maintain readability of the plots we did not include it in the main body of the paper.

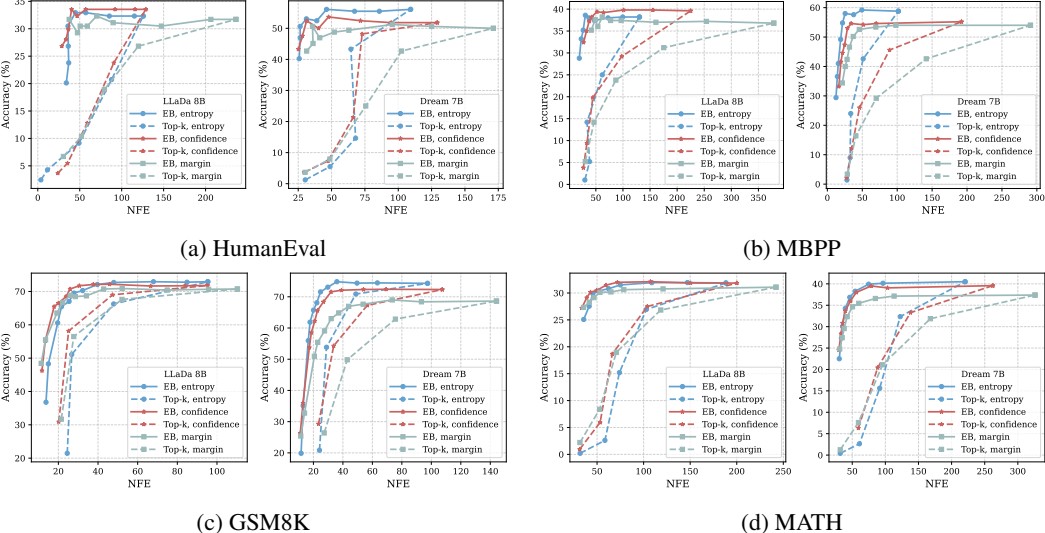

(a) HumanEval

(b) MBPP

(c) GSM8K

(d) MATH

Figure 11: pass@1 accuracy vs. NFE with `generate_until` logic on code and math reasoning tasks.

## D.2 Measuring efficiency of MDMs

In Section 6.1.1 we outline the different ways to measure sampling efficiency and discuss the problems with some of the approaches. We explain why measuring efficiency against full sequence length generation results in apparent high gains (like in Figures 9 and 10), and then propose two ways to better approximate the gains provided by different samplers:

- Unmask with `generate_until` logic

- Unmask semi-auto-regressively with `generate_until` logic

**Effective tokens/step with `generate_until` logic.** In Figure 12 we show accuracy against the effective generation speed of the model quantified via:

$$\text{Effective Tokens/Step} = \frac{\texttt{mean\_answer\_len}}{\texttt{mean\_NFE\_to\_condition}}, \tag{16}$$

where `mean_answer_len` is the average number of tokens from left to right until the `generate_until` answer markers, and `mean_NFE_to_condition` is the number of function evaluations required by the model to generate the answer until both `generate_until` answer markers apear in answer and all tokens before the marker are unmasked. Figure 12 shows that in most cases the effective speed at $\gamma = 0$ or $\text{Top}_1$, is actually less than 1, meaning that the model unmasks tokens that are not used in the final answer, or equivalently unmasks tokens that come after the stopping phrase in the sequence. This observation led us to explore an approach to mitigates this inefficiency via semi-autoregressive generation (Arriola et al., 2025; Nie et al., 2025b).

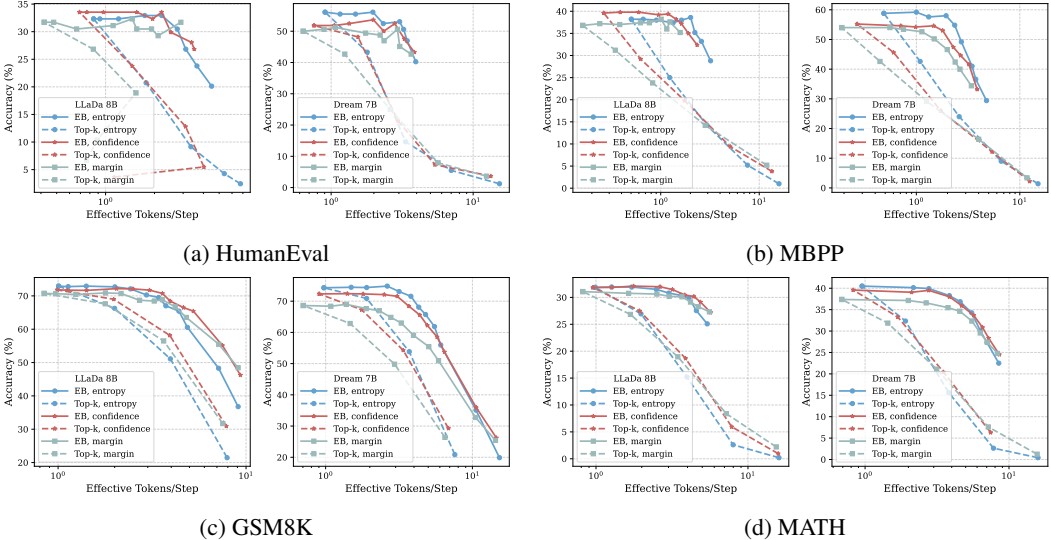

(a) HumanEval

(b) MBPP

(c) GSM8K

(d) MATH

Figure 12: pass@1 accuracy vs. tokens/step on code and math reasoning tasks.

**Semi-autoregressive generation.** Figure 12 supports our claim that mostly in the MBPP benchmark many tokens are generated after the stopping phrase although not all tokens have been unmasked before. In the main body of the paper, in Table 1, we showed an ablation with semi-autoregressive generation for that benchmark with the Dream 7B model. In Table 4 we also add the same ablation with the LLaDa 8B model. We report the same ablation for the GSM8K benchmark in Tables 5 and 6, which show that the gap in efficiency between with and without semi-autoregressive generation is small to non existing aligning with Figure 12 that shows effective token/step of around 1 on this benchmark. That is, on GSM8K it is less likely that the model generates tokens that come after the stopping phrase.

Table 4: NFE and Speed-Ups for LLaDa 8B on the MBPP for various evaluation schemes at roughly same best pass@1. For all configurations in the table the mean answer length is $\sim 60$ tokens.

| | Full `max_gen_len` $= 512$ | | `generate_until` logic | | `generate_until` logic + semi-AR (block_len=64) | | |
|---|---|---|---|---|---|---|---|
| | pass@1 | NFE | Speed-Up | NFE | Speed-Up | pass@1 | NFE | Speed-Up |
| Top-1 | 39.6% | 512 | x 1 | 224.93 | x 1 | 39.4% | 73.31 | x 1 |
| EB, confidence, $\gamma = 0.001$ | 39.8% | 347.28 | x 1.47 | 155.31 | x 1.44 | 39.4% | 60.83 | x 1.21 |
| EB, confidence, $\gamma = 0.1$ | 39.2% | 138.41 | x 3.67 | 64.01 | x 3.51 | 38.8% | 33.20 | x 2.21 |

Table 5: NFE and Speed-Ups for Dream 7B on the GSM8K for various evaluation schemes at roughly same best pass@1. For all configurations in the table the mean answer length is $\sim 93$ tokens.

| | Full `max_gen_len` | | `generate_until` logic | | `generate_until` logic + semi-AR (block_len=64) | | |
|---|---|---|---|---|---|---|---|
| | pass@1 | NFE | Speed-Up | NFE | Speed-Up | pass@1 | NFE | Speed-Up |
| Top-1 | 74.30% | 256 | x 1 | 97.44 | x 1 | 74.90% | 95.23 | x 1 |
| EB, entropy, $\gamma = 0.01$ | 74.37% | 156.29 | x 1.64 | 49.60 | x 1.96 | 75.36% | 48.29 | x 1.97 |
| EB, entropy, $\gamma = 0.1$ | 74.83% | 129.27 | x 1.98 | 35.94 | x 2.71 | 75.36% | 35.60 | x 2.66 |

Table 6: NFE and Speed-Ups for LLaDa 8B on the GSM8K for various evaluation schemes at roughly same best pass@1. For all configurations in the table the mean answer length is $\sim 93$ tokens.

| | Full `max_gen_len` | | `generate_until` logic | | `generate_until` logic + semi-AR (block_len=64) | | |
|---|---|---|---|---|---|---|---|
| | pass@1 | NFE | Speed-Up | NFE | Speed-Up | pass@1 | NFE | Speed-Up |
| Top-1 | 71.79% | 256 | x 1 | 95.19 | x 1 | 71.95% | 93.62 | x 1 |
| EB, confidence, $\gamma = 0.01$ | 71.64% | 185.78 | x 1.36 | 66.57 | x 1.43 | 72.33% | 65.42 | x 1.43 |
| EB, confidence, $\gamma = 0.1$ | 72.17% | 147.48 | x 1.74 | 45.83 | x 2.08 | 72.71% | 45.30 | x 2.07 |

## D.3 Comparison to naive thresholding

Our key observation in this work is that adaptively choosing the number of tokens to unmask in each step can result in a better accuracy-efficiency tradeoff. In this section we compare to naive thresholding approaches that dynamically determine the number of tokens to unmask with a per-token thresholding criterion as opposed to the EB-Sampler which accumulates the sum of entropies.

We test the EB-Sampler against three threshold criteria: entropy, confidence and margin. In fig. 13 we show accuracy vs. Effective Tokens/Step 16. For MBPP we see a very similar profile for EB-Sampler and naive thresholding while for GSM8K, EB-Sampler shows a favorable accuracy-efficiency tradeoff.

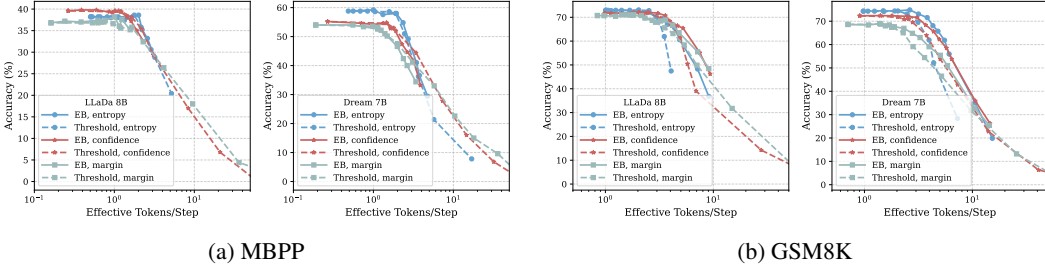

(a) MBPP        (b) GSM8K

Figure 13: Comparison between EB-Sampler and naive thresholding. pass@1 accuracy vs. tokens/step on code and math reasoning tasks.

These results indicate that for some settings, like in MBPP, the model predictions that yield high accuracy are highly confident or equivalently have very low entropy, thus naive thresholding aligns with EB-Sampler accuracy-efficiency profile. On the contrary, for GSM8K, we do observe an non-negligible gap that could be explained by our information theoretic interpretation of the unmasking process.

# E   Societal Impact

As this work introduces a more efficient sampling procedure from existing models, it does not introduce significant societal risks beyond those that already exist.

