# OpenReview forum: "Accelerated Sampling from Masked Diffusion Models via Entropy Bounded Unmasking"
_NeurIPS.cc/2025/Conference — NeurIPS 2025 poster_

### Official Review · Reviewer_6Cof · 2025-06-30

**Clarity:** 4
**Significance:** 4
**Originality:** 3
**Rating:** 6
**Confidence:** 4

**Summary:**

The paper proposes the Entropy Bounded (EB) Sampler for Masked Diffusion Models (MDMs) - a method that unmasks multiple token positions. It hinges on two simple observations: 1. Strong unmasking order is determined by the model; 2. Multiple masked tokens are predicted with high certainty. The EB sampler unmasks multiple such tokens and achieves 2-3x speedup compared to the current sampling algorithms without hurting generation quality.

The paper highlights two known challenges in MDMs:
- Local model error: Since the learned density is approximate, unmasking order matters for sampling, as each step introduces local error.
- Joint correspondence error: Since different positions are sampled independently, the higher the unmask throughput, the more independence assumptions are introduced, lowering quality.

The paper then proposes the Entropy Bounded Sampler, which achieved strong efficiency gains during sampling on various benchmarks.

**Questions:**

- Section 3.2: For joint correspondence error, the standard transformer architecture implies the predictions are not truly independent, so I feel unsure about the independence claim. It would be good to provide some concrete examples to show the failure modes.
- Figure 3: The legends can be improved. It is not clear to me where the current plot shows various k.
- Equation 2: $l$ appeared twice in both the sum and max, which is a little confusing.
- Why does top-k perform quite well on maze and sodoku? What data property is making the gain less prominent than math and reasoning benchmarks?

**Ethical Concerns:**

["NO or VERY MINOR ethics concerns only"]

**Limitations:**

yes

**Quality:**

4

**Strengths And Weaknesses:**

# Quality
- The submission is technically sound, and claims are well supported.

# Clarity
- Overall, the paper is very well written and well organized. The problems are clearly motivated, and the shortcomings of current methods (greedy and top-k) are well presented. The method is intuitive under the authors' motivations. The paper's main plot also highlights the strong performance of the proposed sampler.
- I'm curious about the distribution of the effective k chosen by EB sampler (across various datasets and sampling steps). If it is easy to produce such plots, I encourage the authors to include them in the appendix. Relatedly, it would be awesome to include some qualitative examples where such effective-k varies significantly for the EB sampler. Such plots and examples will benefit future researchers with a better intuition of the model's prediction distribution and further solidify the paper's claim on two types of errors (local model error and joint correspondence error).
- I have raised a few minor clarification questions in the questions section below.

# Significance
- The empirical result is very significant to the field of Masked Diffusion Models. The key appeal of MDMs is that they enable efficient parallel sampling compared to AR models. This result will further solidify this positioning and propel MDMs to be a stronger contender to the existing AR paradigm.
- Figure 5 very strongly highlights the method's significance, showing that the proposed sampler improves the efficiency of current MDMs with drop-in replacement.

# Originality
- The method is simple to understand and has not been applied in the field of MDM before.

---

> ### Author Rebuttal · Authors · 2025-07-30
>
> We thank the reviewer for their valuable feedback. Below, we address the reviewer's questions and concerns.
>
>
>
> > I'm curious about the distribution of the effective k chosen by EB sampler (across various datasets and sampling steps). If it is easy to produce such plots, I encourage the authors to include them in the appendix.
>
> **Answer:**
>
> We thank the reviewer for the suggestion, we will add histograms of effective-k values to the appendix (Unfortunately we cannot attach those here as per the rebuttal text-only responses policy).
>
>
>
> > Relatedly, it would be awesome to include some qualitative examples where such effective-k varies significantly for the EB sampler. Such plots and examples will benefit future researchers with a better intuition of the model's prediction distribution and further solidify the paper's claim on two types of errors (local model error and joint correspondence error).
>
> **Answer:**
>
> We refer the reviewer to the response to reviewer uZV5 where we included an example of the sampling process for an example from the GSM8K benchmark. Again, due to visualization constraints in the rebuttal, we will only add those in the final version of the paper.
>
>
>
> > Section 3.2: For joint correspondence error, the standard transformer architecture implies the predictions are not truly independent, so I feel unsure about the independence claim. It would be good to provide some concrete examples to show the failure modes.
>
> **Answer:**
>
> An MDM produces a marginal distribution per masked token when evaluated at the current state (as described in section XXX). When we unmask multiple tokens simultaneously, the newly unmasked tokens are sampled independently of each other, but importantly conditional on the current state. The joint dependence error comes from failing to account for dependency in this state-conditional sampling.
>
> We will provide qualitative text examples of generation with a large $gamma$ that shows failure modes due to sampling independently conditional on the current state.
>
>
>
> > Figure 3: The legends can be improved. It is not clear to me where the current plot shows various k.
>
> **Answer:**
>
> We will add a comment clarifying that the plot is for 512 tokens generation length. Hence, the points at 512 NFE correspond to $k=1$, at 256 NFE to $k=2$ etc.
>
>
>
>
>
> > Equation 2: appeared twice in both the sum and max, which is a little confusing.
>
> **Answer:**
>
> We will clarify this equation that the sum and max terms are separate, and that the max is not within the sum.
>
>
>
>
>
> > Why does top-k perform quite well on maze and sodoku? What data property is making the gain less prominent than math and reasoning benchmarks?
>
> **Answer:**
>
> One explanation might be that unlike text, where there is more than a single answer to a specific prompt, for Sudoku and mazes, given the initial state, the solution is unique. Hence, the perfect model should be able to solve these problems in a single function evaluation as all tokens are deterministic. We believe that this property allows top-k method to also work relatively well.

---

### Official Review · Reviewer_uZV5 · 2025-07-02

**Clarity:** 2
**Significance:** 2
**Originality:** 3
**Rating:** 4
**Confidence:** 4

**Summary:**

The paper proposes EB-Sampler, a method to accelerate sampling in masked diffusion models (MDMs) for language tasks. The key insight is that partially masked sequences often deterministically constrain multiple unknown tokens, allowing multi-token predictions in a single model call. EB-Sampler exploits this by adaptively unmasking token blocks based on entropy bounds, balancing speed and accuracy. The method yields a 2–3× speedup on code generation and math reasoning tasks without sacrificing performance and generalizes well to structured reasoning problems like mazes and sudoku.

**Questions:**

- Entropy Threshold Sensitivity and Tuning Guidelines: How sensitive is the method's performance to the entropy threshold hyperparameter across different tasks and datasets?
- NFE Control and Predictability: Unlike top-k sampling where the relationship between k and number of function evaluations (NFE) is direct, EB-Sampler's NFE depends on the adaptive entropy-based decisions. How can practitioners predict or control the computational budget when using your method?
- In Figure 9-12, What is the difference between the top-k and margin methods represented by solid and dashed lines in the legend?

**Ethical Concerns:**

["NO or VERY MINOR ethics concerns only"]

**Final Justification:**

I believe that the factual issues regarding this paper have been resolved. The paper offers a new perspective and provides solid evidence for accelerating sampling in diffusion language models, and thus I think it deserves to be accepted. However, as the proposed method represents a relatively incremental improvement over prior work, I did not assign a higher score.

**Limitations:**

Yes

**Quality:**

3

**Strengths And Weaknesses:**

Strengths

- Novel and Well-Motivated Approach: EB-Sampler controls both which and how many tokens to sample using an interpretable entropy bound.
- Simple Implementation: Designed as a drop-in replacement for existing samplers, making it easily adoptable without requiring model retraining or architectural changes.
- Comprehensive Evaluation: Tests on diverse tasks including both natural language generation (code, math) and structured reasoning problems, demonstrating versatility beyond typical language modeling benchmarks.

Weaknesses
- Hyperparameter Sensitivity: The method introduces an entropy threshold as a key hyperparameter, but lacks sufficient analysis of its sensitivity or principled guidelines for tuning across tasks. Unlike top-k methods, which offer a direct and interpretable tradeoff between NFE (number of function evaluations) and k, EB-Sampler makes it harder to control NFE explicitly through hyperparameters.
- Limited Theoretical Analysis:  While the authors decompose the total error into model error and joint dependence error, the theoretical framework only addresses how to reduce the joint dependence error through entropy-bounded unmasking. It remains unclear how their approach mitigates the model error component, which represents the fundamental limitation of the underlying masked diffusion model's predictions, potentially leaving a significant source of error unaddressed.
- Lack of Sampling Examples: The paper focuses heavily on quantitative metrics but provides no actual text samples or sampling process generated by the EB-Sampler.

---

> ### Author Rebuttal · Authors · 2025-07-30
>
> We thank the reviewer for their valuable feedback. Below, we address the reviewer's questions and concerns.
>
>
>
> > Entropy Threshold Sensitivity and Tuning Guidelines: How sensitive is the method's performance to the entropy threshold hyperparameter across different tasks and datasets?
>
> **Answer:**
>
> In all the plots for language modeling the thresholds were: $[0, 0.001, 0.01, 0.1, 0.25, 0.5, 0.75, 1.0, 1.5]$, where performance is roughly maintained until a critical threshold value is reached and performance begins to decline.
>
> For all benchmarks LLaDa has an optimal $\gamma$ between $[0.25,0.5]$ while Dream’s optimal $\gamma$ is in the range of $[0.1,0.25]$, indicating that the threshold is more model dependent rather than task dependent.
>
> In the revised version of the paper we could add plots with $\gamma$ as the x-axis to better present its sensitivity and behaviour across datasets and models if the reviewer finds it to be helpful.
>
>
>
> > NFE Control and Predictability: Unlike top-k sampling where the relationship between k and number of function evaluations (NFE) is direct, EB-Sampler's NFE depends on the adaptive entropy-based decisions. How can practitioners predict or control the computational budget when using your method?
>
> **Answer:**
>
> From the figures, any small value of entropy threshold tends to work very well and performance across the number of steps tends to be flat until the threshold gets large enough. We further observe that the optimal threshold is roughly the same across benchmarks for each model, but the obtained NFE speed up varies. This implies that the required NFE budget may be task dependent, so our suggestion for practitioners is to gradually increase the threshold until performance starts to drop. Given the observed monotonic nature with respect to gamma, tuning this single hyperparameter is a simple process.
>
> Similarly to adaptive solvers for other applications (such as ODEs), the adaptivity enables error control rather than NFE control.
>
>
>
>
>
> > Limited Theoretical Analysis: While the authors decompose the total error into model error and joint dependence error, the theoretical framework only addresses how to reduce the joint dependence error through entropy-bounded unmasking. It remains unclear how their approach mitigates the model error component, which represents the fundamental limitation of the underlying masked diffusion model's predictions, potentially leaving a significant source of error unaddressed.
>
> **Answer:**
>
> Our approach relies upon model error proxies from past research, see Figure 2 in our paper, where it is shown that different model error proxies result in different performance. If improved model error proxies are introduced in the future, they can be easily swapped into EB-Sampler.
>
>
>
>
>
> > In Figure 9-12, What is the difference between the top-k and margin methods represented by solid and dashed lines in the legend?
>
> **Answer:**
> It is a typo in the legend, solid line is always EB-Sampler and dashed lines are Top-k. We will fix it in the revised version.
>
>
>
> > Lack of Sampling Examples: The paper focuses heavily on quantitative metrics but provides no actual text samples or sampling process generated by the EB-Sampler.
>
> **Answer:**
>
> We will add actual text samples in the revised manuscript. Due to text-only replies in the rebuttal policy we write out an example of the sampling process  from the GSM8K dataset, sampling 40 tokens in 17 function evaluations:
>
> ```
> ############ PROMPT ############
> Q: Jean has 30 lollipops. Jean eats 2 of the lollipops. With the remaining lollipops, Jean wants to package 2 lollipops in one bag. How many bags can Jean fill?
>
> A:
> ```
>
> ```
> ############ GENERATION ############
>
>
> --------- Step 0 ----------
> unmasked 1 tokens in this step
> total number of unmasked tokens: 1
> [' ']
> <|mdm_mask|><|mdm_mask|> <|mdm_mask|><|mdm_mask|><|mdm_mask|><|mdm_mask|><|mdm_mask|><|mdm_mask|><|mdm_mask|><|mdm_mask|><|mdm_mask|><|mdm_mask|><|mdm_mask|><|mdm_mask|><|mdm_mask|><|mdm_mask|>
> ---------------------------
>
> --------- Step 1 ----------
> unmasked 1 tokens in this step
> total number of unmasked tokens: 2
> ['0']
> <|mdm_mask|><|mdm_mask|> <|mdm_mask|>0<|mdm_mask|><|mdm_mask|><|mdm_mask|><|mdm_mask|><|mdm_mask|><|mdm_mask|><|mdm_mask|><|mdm_mask|><|mdm_mask|><|mdm_mask|><|mdm_mask|><|mdm_mask|><|mdm_mask|>
> ---------------------------
>
> --------- Step 2 ----------
> unmasked 2 tokens in this step
> total number of unmasked tokens: 4
> ['3', ' ']
> <|mdm_mask|><|mdm_mask|> 30<|mdm_mask|> <|mdm_mask|><|mdm_mask|><|mdm_mask|><|mdm_mask|><|mdm_mask|><|mdm_mask|><|mdm_mask|><|mdm_mask|><|mdm_mask|><|mdm_mask|><|mdm_mask|><|mdm_mask|><|mdm_mask|>
> ---------------------------
>
> --------- Step 3 ----------
> unmasked 4 tokens in this step
> total number of unmasked tokens: 8
> ['2', ' =', ' ', '2']
> <|mdm_mask|><|mdm_mask|> 30<|mdm_mask|> 2 = 2<|mdm_mask|><|mdm_mask|><|mdm_mask|><|mdm_mask|><|mdm_mask|><|mdm_mask|><|mdm_mask|><|mdm_mask|><|mdm_mask|><|mdm_mask|><|mdm_mask|><|mdm_mask|><|mdm_mask|>
> ---------------------------
>
> --------- Step 4 ----------
> unmasked 3 tokens in this step
> total number of unmasked tokens: 11
> ['8', ' l', 'ops']
> <|mdm_mask|><|mdm_mask|> 30<|mdm_mask|> 2 = 28 l<|mdm_mask|>ops<|mdm_mask|><|mdm_mask|><|mdm_mask|><|mdm_mask|><|mdm_mask|><|mdm_mask|><|mdm_mask|><|mdm_mask|><|mdm_mask|><|mdm_mask|><|mdm_mask|>
> ---------------------------
>
> --------- Step 5 ----------
> unmasked 3 tokens in this step
> total number of unmasked tokens: 14
> [' has', ' -', 'ollip']
> <|mdm_mask|> has 30 - 2 = 28 lollipops<|mdm_mask|><|mdm_mask|><|mdm_mask|><|mdm_mask|><|mdm_mask|><|mdm_mask|><|mdm_mask|><|mdm_mask|><|mdm_mask|><|mdm_mask|><|mdm_mask|><|mdm_mask|><|mdm_mask|><|mdm_mask|>
> ---------------------------
>
> --------- Step 6 ----------
> unmasked 1 tokens in this step
> total number of unmasked tokens: 15
> [' Jean']
>  Jean has 30 - 2 = 28 lollipops<|mdm_mask|><|mdm_mask|><|mdm_mask|><|mdm_mask|><|mdm_mask|><|mdm_mask|><|mdm_mask|><|mdm_mask|><|mdm_mask|><|mdm_mask|><|mdm_mask|><|mdm_mask|><|mdm_mask|><|mdm_mask|>
> ---------------------------
>
> --------- Step 7 ----------
> unmasked 1 tokens in this step
> total number of unmasked tokens: 16
> ['.']
>  Jean has 30 - 2 = 28 lollipops<|mdm_mask|>.<|mdm_mask|><|mdm_mask|><|mdm_mask|><|mdm_mask|><|mdm_mask|><|mdm_mask|><|mdm_mask|><|mdm_mask|><|mdm_mask|><|mdm_mask|><|mdm_mask|><|mdm_mask|><|mdm_mask|>
> ---------------------------
>
> --------- Step 8 ----------
> unmasked 1 tokens in this step
> total number of unmasked tokens: 17
> [' left']
>  Jean has 30 - 2 = 28 lollipops left.<|mdm_mask|><|mdm_mask|><|mdm_mask|><|mdm_mask|><|mdm_mask|><|mdm_mask|><|mdm_mask|><|mdm_mask|><|mdm_mask|><|mdm_mask|><|mdm_mask|><|mdm_mask|><|mdm_mask|><|mdm_mask|>
> ---------------------------
>
> --------- Step 9 ----------
> unmasked 1 tokens in this step
> total number of unmasked tokens: 18
> [' ']
>  Jean has 30 - 2 = 28 lollipops left.<|mdm_mask|><|mdm_mask|><|mdm_mask|> <|mdm_mask|><|mdm_mask|><|mdm_mask|><|mdm_mask|><|mdm_mask|><|mdm_mask|><|mdm_mask|><|mdm_mask|><|mdm_mask|><|mdm_mask|><|mdm_mask|>
> ---------------------------
>
> --------- Step 10 ----------
> unmasked 2 tokens in this step
> total number of unmasked tokens: 20
> ['2', ' ']
>  Jean has 30 - 2 = 28 lollipops left.<|mdm_mask|><|mdm_mask|><|mdm_mask|> 2<|mdm_mask|><|mdm_mask|> <|mdm_mask|><|mdm_mask|><|mdm_mask|><|mdm_mask|><|mdm_mask|><|mdm_mask|><|mdm_mask|><|mdm_mask|><|mdm_mask|>
> ---------------------------
>
> --------- Step 11 ----------
> unmasked 7 tokens in this step
> total number of unmasked tokens: 27
> ['8', ' /', '2', ' =', ' ', '1', '4']
>  Jean has 30 - 2 = 28 lollipops left.<|mdm_mask|><|mdm_mask|><|mdm_mask|> 28 / 2 = 14<|mdm_mask|><|mdm_mask|><|mdm_mask|><|mdm_mask|><|mdm_mask|><|mdm_mask|><|mdm_mask|><|mdm_mask|><|mdm_mask|><|mdm_mask|>
> ---------------------------
>
> --------- Step 12 ----------
> unmasked 2 tokens in this step
> total number of unmasked tokens: 29
> [' can', ' bags']
>  Jean has 30 - 2 = 28 lollipops left.<|mdm_mask|> can<|mdm_mask|> 28 / 2 = 14 bags<|mdm_mask|><|mdm_mask|><|mdm_mask|><|mdm_mask|><|mdm_mask|><|mdm_mask|><|mdm_mask|><|mdm_mask|><|mdm_mask|><|mdm_mask|><|mdm_mask|>
> ---------------------------
>
> --------- Step 13 ----------
> unmasked 1 tokens in this step
> total number of unmasked tokens: 30
> [' Jean']
>  Jean has 30 - 2 = 28 lollipops left. Jean can<|mdm_mask|> 28 / 2 = 14 bags<|mdm_mask|><|mdm_mask|><|mdm_mask|><|mdm_mask|><|mdm_mask|><|mdm_mask|><|mdm_mask|><|mdm_mask|><|mdm_mask|><|mdm_mask|><|mdm_mask|>
> ---------------------------
>
> --------- Step 14 ----------
> unmasked 1 tokens in this step
> total number of unmasked tokens: 31
> [' answer']
>  Jean has 30 - 2 = 28 lollipops left. Jean can<|mdm_mask|> 28 / 2 = 14 bags<|mdm_mask|><|mdm_mask|> answer<|mdm_mask|><|mdm_mask|><|mdm_mask|><|mdm_mask|><|mdm_mask|><|mdm_mask|><|mdm_mask|><|mdm_mask|><|mdm_mask|>
> ---------------------------
>
> --------- Step 15 ----------
> unmasked 8 tokens in this step
> total number of unmasked tokens: 39
> ['.', ' The', ' is', ' ', '1', '4', '.', '<|endoftext|>']
>  Jean has 30 - 2 = 28 lollipops left. Jean can<|mdm_mask|> 28 / 2 = 14 bags. The answer is 14.<|endoftext|><|mdm_mask|><|mdm_mask|><|mdm_mask|><|mdm_mask|><|mdm_mask|><|mdm_mask|><|mdm_mask|><|mdm_mask|><|mdm_mask|>
> ---------------------------
>
> --------- Step 16 ----------
> unmasked 1 tokens in this step
> total number of unmasked tokens: 40
> [' make']
>  Jean has 30 - 2 = 28 lollipops left. Jean can make 28 / 2 = 14 bags. The answer is 14.<|endoftext|><|mdm_mask|><|mdm_mask|><|mdm_mask|><|mdm_mask|><|mdm_mask|><|mdm_mask|><|mdm_mask|><|mdm_mask|><|mdm_mask|><|mdm_mask|>
> ```

---

> > ### Comment · Reviewer_uZV5 · 2025-08-04
> >
> > Thank you for your response. After carefully considering your rebuttal along with the feedback from other reviewers, I have decided to maintain my score as Borderline Accept.

---

### Official Review · Reviewer_UqMT · 2025-07-03

**Clarity:** 3
**Significance:** 2
**Originality:** 2
**Rating:** 4
**Confidence:** 4

**Summary:**

This paper proposed Entropy-Bounded (EB) sampler for Masked (Discrete) Diffusion Models (MDMs) as an potential improvement compared to existing samplers for MDMs. The main idea is motivated from the top-$k$ sampling method, which first picks some model error proxy (Examples include entropy or marginals associated with the factorized conditional distributions) as some fixed metric, chooses the top-$k$ tokens with the lowest values of the fixed metric and unmasks the $k$ tokens simultaneously. However, one major drawback of the top-$k$ sampling method is that the joint dependency error incurred by the $k$ tokens increases as $k$ becomes larger. In order to resolve such issue, the authors proposed the EB sampler, which controls the size $k$ adaptively by setting a threshold on the joint dependency error and select the subset of tokens to be unmasked in a greedy way. In order to justify the validity of the proposed EB sampler, the authors provided both theoretical justifications and a complete set of numerical experiments.

**Questions:**

For the theoretical part, the reviewer is wondering whether it is possible to perform some analysis on the proposed EB sampler to see how the final discrepancy between the target distribution and the distribution of the generated sample scale with respect to the discretized timestep. The authors might consider referring to the stochastic integral framework proposed and used in [1,2] to see how the techniques developed in these two papers may be applied to analyze the proposed EB sampler to better understand the its effectiveness theoretically. Other references that the reviewer thinks might be helpful are [3,4,5]. Specifically, [4] provides the first theoretical analysis of masked discrete diffusion models to the best of the reviewer's knowledge. Would it be possible for the authors to at least cite these papers and briefly comment on how the EB sampler might be analyzed theoretically while revising the manuscript?

For the empirical part, on the other hand, the reviewer thinks it might still be necessary to include extra baselines like Autoregressive Models (ARMs) in Figure 5,6 and 7 as they are more commonly used nowadays in the tasks included in this paper. Would it be possible for the authors to add ARM as a baseline method for comparison?

Finally, there are also a few miscellaneous problems regarding the presentation and typos of the manuscript. For instance, both Figure 10 and 11 contain two plots for the same method "Top-$k$, margin" - may the reviewer ask whether this is a typo or not? Also, the notation "arg Top2" used in line 113 might need to have some text-based explanation as this is not a commonly used notation.

**Ethical Concerns:**

["NO or VERY MINOR ethics concerns only"]

**Final Justification:**

The reviewer has checked all of the reviews and the authors' responses. Overall, most issues have been resolved and the reviewer is satisfied with the additional experimental results that the authors added. Hence, the reviewer is recommending "Borderline Accept" by raising the score from 3 to 4 (provided that the authors will include all changes mentioned in the reviews and rebuttals for the camera-ready version of the paper).

**Limitations:**

The authors briefly commented on what future work might be performed to further improve the proposed EB sampler, which the reviewer thinks is sufficient for addressing potential limitations of the work. One other possible issue that the authors might also consider resolving is the gap between the objective (9) proposed for thresholding and the true joint mutual information within the subset of sampled tokens.

References:

[1] Ren, Y., Chen, H., Rotskoff, G. M., and Ying, L. How discrete and continuous diffusion meet: Comprehensive analysis of discrete diffusion models via a stochastic integral framework. In The Thirteenth International Conference on Learning Representations (ICLR), 2025

[2] Ren, Y., Chen, H., Zhu, Y., Guo, W., Chen, Y., Rotskoff, G. M., Tao, M., and Ying, L. Fast solvers for discrete diffusion models: Theory and applications of high-order algorithms. arXiv preprint arXiv:2502.00234, 2025a

[3] Zhang, Z., Chen, Z., & Gu, Q. (2024). Convergence of score-based discrete diffusion models: A discrete-time analysis. In The Thirteenth International Conference on Learning Representations (ICLR), 2025

[4] Liang, Y., Huang, R., Lai, L., Shroff, N., & Liang, Y. (2025). Absorb and Converge: Provable Convergence Guarantee for Absorbing Discrete Diffusion Models. arXiv preprint arXiv:2506.02318.

[5] Chen, H., & Ying, L. (2024). Convergence analysis of discrete diffusion model: Exact implementation through uniformization. arXiv preprint arXiv:2402.08095.

**Quality:**

3

**Strengths And Weaknesses:**

Pros: The reviewer finds that the manuscript is presented in a clear and straightforward way. The proposed method is simple and effective with a clear intuition. Large-scale experiments on code generation, math reasoning and solving logic puzzles are provided to justify the effectiveness of the proposed method.

Cons: The overall novelty seems to be a little bit limited as it is a simple modification of the existing top-$k$ method. Theoretically, the reviewer thinks the analysis provided in this paper could possibly be enhanced by using tools developed in recent/concurrent work on the theoretical properties of (masked) discrete diffusion models. Empirically, though the reviewer totally understands that the main focus of this paper is to improve the top-$k$ sampling method, it might be still be worthwhile to add other more commonly used models and methods as baselines. The weaknesses will be further elaborated in the "Questions" section below.

---

> ### Author Rebuttal · Authors · 2025-07-30
>
> We thank the reviewer for their valuable feedback. Below, we address the reviewer's questions and concerns.
>
>
>
> > The overall novelty seems to be a little bit limited as it is a simple modification of the existing top-$k$ method.
>
> **Answer:**
>
> While the EB-Sampler is a simple modification to existing top-$k$ methods, it is **new** in MDM sampling literature and it provides **unprecedented strong efficiency improvements at zero cost**. Simplicity that brings strong improvements should be considered as a strong suit rather than a disadvantage.
>
> We would also like to highlight the contribution of the mathematical framework we introduce that covers many other adaptive multitoken sampling schemes for (time independent) masked discrete diffusion, extending upon the adaptive MDM inference from [Kim et. al. (2025)]([https://arxiv.org/abs/2502.06768).
>
>
>
> > For the theoretical part, the reviewer is wondering whether it is possible to perform some analysis on the proposed EB sampler to see how the final discrepancy between the target distribution and the distribution of the generated sample scale with respect to the discretized timestep. The authors might consider referring to the stochastic integral framework proposed and used in [1,2] to see how the techniques developed in these two papers may be applied to analyze the proposed EB sampler to better understand the its effectiveness theoretically. Other references that the reviewer thinks might be helpful are [3,4,5]. Specifically, [4] provides the first theoretical analysis of masked discrete diffusion models to the best of the reviewer's knowledge. Would it be possible for the authors to at least cite these papers and briefly comment on how the EB sampler might be analyzed theoretically while revising the manuscript?
>
> **Answer:**
>
> We appreciate the suggestions from the reviewer and will add citations to these papers, where we note [2] is already cited in related work.
>
> We quantify the discrepancy between the target distribution and the model distribution via the KL divergence in Section 5, proving the decomposition into model error and joint dependence error in Appendix A. Unlike more general continuous-time discrete diffusion, we have no explicit time discretization error from our sampling scheme. Our MDM sampling dispenses with time entirely as the optimal solution for MDM predictions are clean data conditionals, and time-independent, as described in Section 2.2 (see [Ou et. al. (2025)](https://arxiv.org/abs/2406.03736), [Zheng et. al. (2024)](https://arxiv.org/abs/2409.02908)). Naturally many MDM’s, including LLaDa and Dream, are parameterized without time given this optimal solution. That time is not required for MDM is embodied within our framework of adaptive multi-token sampling in Section 5 that considers state transitions via unmasking as opposed to following the state of a discrete diffusion process over time. This framework is expressive enough to contain existing (top-k) as well as novel adaptive multi-token MDM sampling schemes, including EB-Sampler, while avoiding heavyweight mathematical machinery needed for more general continuous-time discrete diffusion.
>
> EB-Sampler only bounds the KL divergence in this framework if there is zero model error amongst the unmasked tokens. A theoretical understanding of how EB-Sampler would behave under inaccurate model predictions could likely benefit from the more advanced techniques introduced in these references.
>
>
>
> > For the empirical part, on the other hand, the reviewer thinks it might still be necessary to include extra baselines like Autoregressive Models (ARMs) in Figure 5,6 and 7 as they are more commonly used nowadays in the tasks included in this paper. Would it be possible for the authors to add ARM as a baseline method for comparison?
>
> **Answer:**
>
> Our paper focuses on accelerating MDM sampling for SOTA large language diffusion models like [LLaDa](https://arxiv.org/abs/2502.09992) and [Dream](https://hkunlp.github.io/blog/2025/dream/) that have already been shown to be competitive with similarly sized ARM models in terms of performance on these tasks.
>
> Would adding a point to the figures showing performance of similar sized ARMs sampled with full-NFE next-token prediction be helpful? If so, we will add it in the revised version.
>
>
>
> > Finally, there are also a few miscellaneous problems regarding the presentation and typos of the manuscript. For instance, both Figure 10 and 11 contain two plots for the same method "Top- , margin" - may the reviewer ask whether this is a typo or not? Also, the notation "arg Top2" used in line 113 might need to have some text-based explanation as this is not a commonly used notation.
>
> **Answer:**
>
> Regarding Figures 10,11,12 - We thank the reviewer for pointing this out. It is indeed a typo in the legend, we will fix it in the revised manuscript. To clarify, solid lines are always EB-Sampler and dashed lines are Top-k.
>
> We will also add an explanation for the $\arg \mathrm{Top}_k$ notation.

---

> > ### Comment · Reviewer_UqMT · 2025-08-05
> > **Response to the authors**
> >
> > The reviewer would like to thank the authors for the detailed response, which has addressed most of the reviewer's concerns. Given that theoretical analysis appears to be a primary limitation of the manuscript, the authors should probably consider citing the articles [1,3,4,5] listed in the references above and briefly discuss how one may theoretically analyze the EB-sampler as future work. Moreover, it might also be beneficial to compare Top-K method here with the "Best-of-N" sampling method commonly used in (masked) discrete diffusion models and LLMs, as they seem to be the same (type of) method. One may refer to [6] to see a version of the "Best-of-N" sampling method for the setting of diffusion models.
> >
> > Also, the reviewer is satisfied with the authors' response to other reviewers' questions. The authors are strongly encouraged to add the extra experimental results, such as the text examples, in the camera-ready version of the manuscript. Overall, the reviewer will recommend accepting the paper by raising the score from 3 to 4.
> >
> > Added References:
> >
> > [6] Uehara, Masatoshi, et al. "Inference-time alignment in diffusion models with reward-guided generation: Tutorial and review." arXiv preprint arXiv:2501.09685 (2025).

---

### Official Review · Reviewer_RBWG · 2025-07-06

**Clarity:** 3
**Significance:** 3
**Originality:** 2
**Rating:** 4
**Confidence:** 3

**Summary:**

The paper proposes a technique for adaptively unmasking tokens (instead of fixed K tokens for some hyperparameter K) for a masked diffusion model, based on an entropy bound for the unsampled tokens. It provides results on LLADA 8B and Dream 7B models, which demonstrate significantly robust performance at small NFE as compared to Top-K methods.

**Questions:**

1. How does the optimal threshold gamma for the method in the paper scale with the vocabulary size or the generation length? Also, should it depend on some average entropy level of the predictions?

**Ethical Concerns:**

["NO or VERY MINOR ethics concerns only"]

**Limitations:**

I do not think that the work has any major limitations, apart from missing the baselines as stated before.

**Quality:**

4

**Strengths And Weaknesses:**

The main strength of the work is providing a novel adaptive token unmasking scheme, which can choose the number of tokens to unmask at a given time. It shows impressive results on various models such as LLADA and Dream in terms of performance degradation with NFE.

I do not think the paper has significant weaknesses. But I think the paper would benefit by including some baselines, based on using thresholds on metrics such as confidence, entropy and margin. The baseline could be the following:

1. Unmask all tokens having confidence greater than a threshold
2. Unmask all tokens having entropy less than a threshold
3. Unmask all tokens having margin greater than a threshold

I think the contribution of the paper is important, but these baselines would be useful for understanding why the method works.

---

> ### Author Rebuttal · Authors · 2025-07-30
>
> We thank the reviewer for their valuable feedback. Below, we address the reviewer's questions and concerns.
>
>
> > But I think the paper would benefit by including some baselines, based on using thresholds on metrics such as confidence, entropy and margin. The baseline could be the following:
> >
> > 1. Unmask all tokens having confidence greater than a threshold
> > 2. Unmask all tokens having entropy less than a threshold
> > 3. Unmask all tokens having margin greater than a threshold
> >
> > I think the contribution of the paper is important, but these baselines would be useful for understanding why the method works.
>
> **Answer:**
>
> We appreciate the suggestion to explore additional members of the $\phi$ family (eq. (5) in paper).  This threshold strategy, specifically for confidence, was introduced in concurrent research [Wu et. al. (2025)](https://arxiv.org/abs/2505.22618). As described in our paper, these alternative $\phi$'s can also leverage pre-trained masked diffusion models based on our multi-token adaptive sampling framework. We found that the suggested alternative $\phi$ performed similarly to EB-Sampler on MBPP but worse on GSM8K. In particular, for similar performance, EB-Sampler enables sampling more tokens on GSM8K.
>
> For the different models we report `NFE / Accuracy%` in the table below for the optimal $\gamma$. For LLaDa we use `confidence` and for Dream we use `entropy` (see Tables 1,4,5,6 in paper).
>
> | Benchmark                 | Model |   Top$_1$   |   Threshold   |  EB-Sampler   |
> | ------------------------- | ----- | :---------: | :-----------: | :-----------: |
> | MBPP (`max_gen_len`=512)  | LLADa | 512 / 39.6% | 97.2 / 39.4%  | 109.2 / 39.4% |
> |                           | Dream | 512 / 58.8% | 84.7 / 58.0%  | 85.3 / 58.0%  |
> | GSM8K (`max_gen_len`=256) | LLaDa | 256 / 71.8% | 138.2 / 71.9% | 115.4 / 71.7% |
> |                           | Dream | 256 / 74.3% | 150.1 / 74.4% | 129.3 / 74.8% |
>
> In the revised manuscript we will incorporate these results as plots for all choices of error proxies for both models.
>
> An interpretation of these results is that, even if model error is non-existent, an absolute threshold on token entropies/confidence could allow for an arbitrary amount of error to be introduced in sampling. Thresholding the entropy sum instead controls the overall error. This difference is more relevant when there is more variation in the token entropies. If all entropies were roughly the same, then an absolute threshold would behave similarly to an entropy sum threshold, which we observe does not occur on GSM8K.
>
>
>
>
>
> > How does the optimal threshold gamma for the method in the paper scale with the vocabulary size or the generation length?
>
> **Answer:**
>
> *Generation length:* We experimented with semi-autoregressive generation with varying block lengths for MBPP and GSM8K for both LLaDA and Dream. Results are in the tables below.
>
> We observe that LLaDa maintains a stable optimal $\gamma$ values for different block sizes, while Dream seems to be more sensitive, requiring lower $\gamma$ values for smaller blocks on MBPP but higher $\gamma$ values on GSM8K.
>
> We believe that the major factors affecting the optimal threshold values are the model itself and the task when blocks are very short. For large blocks such as 256 or 512, the optimal threshold seems stable across datasets for each of the models.
>
>
>
>
>
> MBPP, LLaDa, semi-autoregressive generation
>
> | $\gamma$ / block_len | 16   | 32   | 64   | 128  | 512  |
> | -------------------- | ---- | ---- | ---- | ---- | ---- |
> | 0                    | 39.6 | 38.4 | 39.4 | 39.4 | 39.6 |
> | 0.01                 | 39.4 | 38.2 | 39.4 | 39.6 | 39.8 |
> | 0.1                  | 39.6 | 38.0   | 38.8 | 39.2 | 39.2 |
> | 0.25                 | 39.2 | 38.0   | 39.0   | 39.2 | 39.4 |
> | 0.5                  | 36.8 | 36.8 | 37.6 | 38.2 | 38.2 |
>
> MBPP, Dream, semi-autoregressive generation
>
> | $\gamma$ / block_len | 16   | 32   | 64   | 128  | 512  |
> | -------------------- | ---- | ---- | ---- | ---- | ---- |
> | 0                    | 57.2 | 58.0   | 58.6 | 59.0   | 58.8 |
> | 0.01                 | 56.6 | 57.6 | 57.2 | 57.8 | 57.6 |
> | 0.1                  | 54.4 | 56.2 | 58.6 | 58.2 | 58.0   |
> | 0.25                 | 50.6 | 53.8 | 55.2 | 55.0   | 54.8 |
> | 0.5                  | 46.0   | 48.6 | 49.4 | 49.2 | 49.2 |
>
>
>
> GSM8K, LLaDa, semi-autoregressive generation
>
> | $\gamma$ / block_len | 16   | 32   | 64   | 128  | 256  |
> | -------------------- | ---- | ---- | ---- | ---- | ---- |
> | 0                    | 73.1 | 72.6 | 71.9 | 71.8 | 71.8 |
> | 0.01                 | 73.1 | 72.6 | 72.3 | 71.7 | 71.6 |
> | 0.1                  | 72.8 | 72.4 | 72.7 | 72.0 | 72.2 |
> | 0.25                 | 73.0 | 73.1 | 72.9 | 72.2 | 72.1 |
> | 0.5                  | 73.1 | 72.2 | 71.9 | 71.9 | 71.7 |
> | 0.75                 | 71.0 | 70.7 | 70.1 | 70.6 | 70.7 |
>
> GSM8K, Dream, semi-autoregressive generation
>
> | $\gamma$ / block_len | 16   | 32   | 64   | 128  | 256  |
> | -------------------- | ---- | ---- | ---- | ---- | ---- |
> | 0                    | 75.7 | 75.3 | 74.9 | 74.7 | 74.3 |
> | 0.01                 | 75.3 | 74.8 | 75.4 | 75.0 | 74.4 |
> | 0.1                  | 74.1 | 74.6 | 75.4 | 75.2 | 74.8 |
> | 0.25                 | 74.8 | 74.1 | 74.2 | 73.9 | 73.1 |
> | 0.5                  | 71.3 | 71.8 | 71.9 | 72.0 | 71.6 |
>
>
>
> *Vocabulary size:* For the evaluated language MDMs in the paper, we do observe a difference in the optimal $\gamma$ value (LLaDa - $\gamma=0.25$, Dream - $\gamma=0.1$). Although LLaDa and Dream have different vocabulary sizes (126k and 151k, respectively), this alone is insufficient to determine the effect of vocabulary size. These models were trained differently, using distinct data and context lengths. A computationally intensive experiment involving training multiple models in a controlled environment would be necessary to isolate the influence of vocabulary size on the optimal $\gamma$.
>
>
>
>
>
> > Also, should it depend on some average entropy level of the predictions?
>
> **Answer:**
>
> We do observe that for the two language MDMs used in the paper the empirically observed optimal $\gamma$ is different: $\gamma=0.25$ for LLaDa and $\gamma=0.1$ for Dream, but they are consistent across benchmarks. The reviewer is correct though that potentially one could improve our results further by learning state-dependent entropy thresholds for EB-Sampler or more generally state-dependent $\phi$ functions as we discuss in the conclusions section. Tuning more than one parameter would require more substantive effort though, and we leave learning possibly even stronger adaptive unmasking schemes as extensions for future research.

---

### Note · Authors · 2025-08-15

We sincerely thank the reviewers for their feedback. We appreciate the recognition in the strong performance of our Entropy Bounded (EB) Sampler as a simple yet effective adaptive multi-token unmasking method that achieves substantial speedups without sacrificing accuracy.

Per the discussion period, we received replies from (i) reviewer UqMT who replied they will raise their score from 3 to 4; and (ii) reviewer uZV5 who replied they will keep their initial score at 4.

Key points we would like to highlight:

- The EB-Sampler introduces a novel modification to existing top-k methods in the masked diffusion model literature, delivering **unprecedented efficiency improvements at zero cost** and backed with theoretical justification. Its simplicity is a key strength that enables strong empirical gains and practical adoption ([see response to reviewer UqMT](https://openreview.net/forum?id=WBcBhT1NKO&noteId=VktiDRUgkm)).
- We clarified theoretical aspects of our work which addresses the simple masked generative modeling framework which is independent of time. We will add citations to recent relevant works to contextualize our framework and acknowledge possible extensions of our analysis ([see response to reviewer UqMT](https://openreview.net/forum?id=WBcBhT1NKO&noteId=VktiDRUgkm)).
- We have addressed suggestions by including additional baselines, showing that EB-Sampler performs favorably compared to these alternatives ([see response to reviewer RBWG](https://openreview.net/forum?id=WBcBhT1NKO&noteId=G3muuFiyel)).
- We commit to adding qualitative examples, sampling process illustrations, and histograms of effective token unmasking counts in the final version to aid intuition and reproducibility ([see response to reviewer uZV5](https://openreview.net/forum?id=WBcBhT1NKO&noteId=fvEPH60OXV)).

Our method offers a straightforward, effective way to accelerate masked diffusion model sampling without loss of accuracy, strengthening the case for these models in language and reasoning tasks. We hope the ACs find our responses and planned improvements satisfactory.

---

### Decision · Program_Chairs · 2025-09-17

**Decision:**

Accept (poster)

**Comment:**

This paper proposes the entropy-bound sampler, a method for dynamically determining parallel unmasking bandwidth in masked diffusion models. All reviewers recommend acceptance -- although some concerns still remain after rebuttal, the reviewers have reached consensus that the paper should be accepted provided authors incorporate all promised revisions from the rebuttal into the camera-ready version.

A critical finding from the rebuttal is that the method's gains are primarily due to its dynamic unmasking bandwidth, while the empirical comparisons are mostly conducted against "top-k" methods with a fixed k. Additional experiments requested by Reviewer RBWG shows that applying threshold on confidence (which also introduces a dynamic k) achieved similar results.

The authors are strongly encouraged to compare to other baselines with a dynamic k in the MDM literature such as the cosine discretization approach (Shi et al., 2024) and the margin-based thresholding (Kim et al., 2025), which would provide a more comprehensive picture of the relative gains of the EB sampler.